# Dual domain recognition determines SARS-CoV-2 PLpro selectivity for human ISG15 and K48-linked di-ubiquitin

Pawel M. Wydorski [1,2,9], Jerzy Osipiuk [3,4,9], Benjamin T. Lanham [5,9], Christine Tesar [3,4], Michael Endres [3,4], Elizabeth Engle [5], Robert Jedrzejczak [3,4], Vishruth Mullapudi [2], Karolina Michalska [3,4], Krzysztof Fidelis [6], David Fushman [5] ✉, Andrzej Joachimiak [3,4,7] ✉ & Lukasz A. Joachimiak [2,8] ✉

The Papain-like protease (PLpro) is a domain of a multi-functional, non-structural protein 3 of coronaviruses. PLpro cleaves viral polyproteins and posttranslational conjugates with poly-ubiquitin and protective ISG15, composed of two ubiquitin-like (UBL) domains. Across coronaviruses, PLpro showed divergent selectivity for recognition and cleavage of posttranslational conjugates despite sequence conservation. We show that SARS-CoV-2 PLpro binds human ISG15 and K48-linked di-ubiquitin (K48-Ub$_2$) with nanomolar affinity and detect alternate weaker-binding modes. Crystal structures of untethered PLpro complexes with ISG15 and K48-Ub$_2$ combined with solution NMR and cross-linking mass spectrometry revealed how the two domains of ISG15 or K48-Ub$_2$ are differently utilized in interactions with PLpro. Analysis of protein interface energetics predicted differential binding stabilities of the two UBL/Ub domains that were validated experimentally. We emphasize how substrate recognition can be tuned to cleave specifically ISG15 or K48-Ub$_2$ modifications while retaining capacity to cleave mono-Ub conjugates. These results highlight alternative druggable surfaces that would inhibit PLpro function.

The COVID-19 pandemic is caused by Severe Acute Respiratory Syndrome Coronavirus 2 (SARS-CoV-2) from the *Coronaviridae* family[1,2], a spherical, enveloped, non-segmented, (+) sense RNA virion with ~30 kbs genome. The RNA is used for the synthesis of two polyproteins (Pp1a/Pp1ab), which are processed by two viral proteases: papain-like protease (PLpro) and 3C-like protease (Mpro). PLpro is a domain of

non-structural protein 3 (Nsp3). It cleaves three sites in SARS-CoV-2 polyproteins yielding Nsp1, Nsp2, and Nsp3. The LXGG↓XX motif found in the polyproteins is essential for protease recognition and cleavage. PLpro has been shown to have additional functions, including the best characterized deubiquitinating[3–5] and deISGylating activities[6,7], occurring by the cleavage of conjugates having the LRGG

[1]Molecular Biophysics Graduate Program, University of Texas Southwestern Medical Center, Dallas, TX 75390, USA. [2]Center for Alzheimer's and Neurodegenerative Diseases, Peter O'Donnell Jr. Brain Institute, University of Texas Southwestern Medical Center, Dallas, TX 75390, USA. [3]Center for Structural Biology of Infectious Diseases, Consortium for Advanced Science and Engineering, University of Chicago, Chicago, IL 60667, USA. [4]Structural Biology Center, X-ray Science Division, Argonne National Laboratory, Lemont, IL 60439, USA. [5]Department of Chemistry and Biochemistry, Center for Biomolecular Structure and Organization, University of Maryland, College Park, MD 20742, USA. [6]Protein Structure Prediction Center, Genome and Biomedical Sciences Facilities, University of California, Davis, CA 95616, USA. [7]Department of Biochemistry and Molecular Biology, University of Chicago, Chicago, IL 60367, USA. [8]Department of Biochemistry, University of Texas Southwestern Medical Center, Dallas, TX 75390, USA. [9]These authors contributed equally: Pawel M. Wydorski, Jerzy Osipiuk, Benjamin T. Lanham. ✉e-mail: fushman@umd.edu; andrzejj@anl.gov; Lukasz.Joachimiak@utsouthwestern.edu

sequence motif found at the C-terminus of ubiquitin (Ub) and ISG15. These activities dysregulate Ub- and ISG15-dependent pathways which play important roles in protein degradation, vesicular trafficking, inflammatory, anti-pathogen responses, and homeostasis[7,8]. Interestingly, the SARS-CoV-1 PLpro has a strong preference for hydrolysis of K48-linked polyUb conjugates, in contrast to the MERS-CoV enzyme that acts as a general, broad-specificity deubiquitinase (DUB)[9]. Removing these modifications disturbs interferon (IFN) expression and blocks NF-kappaB signaling[9], and cleaving off ISG15 from STAT induces up-regulation of TGF-β1[10]. Deubiquitination and deISGylation were shown to be utilized by a broad family of viruses that include Coronaviruses, Hepadnaviruses, Nairoviruses, Arteriviruses, Picornaviruses, and Aphthoviruses[11]. Some other PLpro functions involve direct cleavage of host proteins influencing wide-ranging processes from blood coagulation to nuclear transport[12–14]. PLpro may also play roles beyond its proteolytic activity[5], illustrating its diverse and complex functions[15]. The SARS-CoV-2 PLpro (PLpro$^{CoV-2}$) sequence and structural fold are conserved among SARS-CoV-1 (83% identical), MERS-CoV (30% identical), and other coronaviruses. Despite low sequence identities (~10%)[6], PLpros also share common structural architecture and catalytic site with the human ubiquitin-specific proteases (USPs), one of the five distinct deubiquitinating enzyme (DUB) families. Interestingly, one of these USPs (USP18) is specific for cleaving off ISG15 conjugates in humans and other vertebrates, and PLpro$^{CoV-2}$ deISGylation activity may potentiate USP18 regulatory function[16].

The PLpro$^{CoV-2}$ differs significantly from Mpro, which was shown to recognize linear sequence motif[17], as it encodes proteolytic activities that control viral polyprotein cleavage but also is processing conjugates of host and viral proteins, including polyUb and ISG15. Therefore, PLpro$^{CoV-2}$, in addition to recognizing the LXGG↓XX linear motif, can also specifically interact with protein surfaces presented in three-dimensional structures and, using this mechanism, can recognize and select different substrates. Decoupling the basal polyprotein cleavage activity from those required to process polyUb or ISG15 is important for understanding their role in viral pathogenesis, how these activities influence the interaction of the virus with the host immune system, and how changes in these interactions modify host antiviral responses and influence disease outcomes.

Ubiquitination is an essential posttranslational modification (PTM) engaged in multiple functions in humans, including signaling and proteasome-dependent protein degradation[18]. The modification is mediated by the Ub-conjugating system and could be reversed by DUBs[7]. *ISG15* is an IFNα-stimulated gene that is a critical component of the antiviral response[19,20]. ISG15 is processed and subsequently activated in a manner similar to Ub using interferon-induced factors that follow the ubiquitination-like E1, E2, and E3 enzyme cascade to mediate co-translational ISGylation—an addition of ISG15, via its C-terminal LRGG motif, to substrate lysine residues[21]. It is not precisely clear how ISG15 interferes with viral processes but it is believed that tagging newly translated viral proteins with ISG15 sterically prevents their folding, assembly or interactions[22]. The level of ISGylation is controlled by interferon and USP18[16]. The free unconjugated ISG15 form can exist intracellularly or be secreted to function as a cytokine, linked to the induction of a cytokine storm[23,24]. Removal of Ub and ISG15 conjugates from specific substrates in host cells may have a diverse impact on numerous cellular processes and specifically may disrupt the host response to viral infection[19,20,25]. PLpro can recognize both appendages and cleave them off as they share a PLpro recognition motifs at their C-termini. PolyUb and ISG15 have common other structural features: ISG15 comprises two Ub-like (UBL) domains and mimics a head-to-tail linked di-ubiquitin (Ub$_2$). While K48-linked Ub$_2$ (K48-Ub$_2$) and ISG15 are similar both in sequence and fold, the topologies of how the two domains are linked are distinct. How PLpro discriminates between different ubiquitin linkage types and specifically between Ub$_2$ and

ISG15 substrates is still unknown. Understanding how PLpro discriminates between different substrates will help uncover how these additional proteolytic activities contribute to viral pathogenesis.

Recently published work suggested that mutations in PLpro$^{CoV-1}$ to PLpro$^{CoV-2}$ changed its binding preference from K48-Ub$_2$ to human ISG15 (hISG15)[26,27]. Inspired and attracted by these results, we investigated the interaction of PLpro$^{CoV-2}$ with mono-, di-, and tri-ubiquitins (Ub$_1$, Ub$_2$, and Ub$_3$) and hISG15. We employed complementary biochemical, structural X-ray, NMR, mutagenesis, and computational approaches, to understand how PLpro$^{CoV-2}$ can differentiate between linear recognition motifs, Ub$_1$, K48-Ub$_2$, and hISG15 substrates. We find that PLpro$^{CoV-2}$ binds both hISG15 and K48-Ub$_2$ with high and similar affinity but shows weaker interactions with Ub$_1$. We also find lower affinity alternate binding modes for hISG15, K48-Ub$_2$, and Ub$_1$, which can be explained by non-stoichiometric binding modes and sequence preference outside the conserved recognition motif. We also observe how amino acid substitutions in the first position X following G in the LXGG↓(X) motif impact peptide cleavage. To reveal details of substrate recognition, we determined structures of non-covalent complexes of PLpro$^{CoV-2}$ (single C111S and double C111S, D286N proteolytically inactive mutants) with hISG15 and K48-Ub$_2$. Our crystal structures of non-modified, complete complexes uncover that hISG15 binding is determined by the recognition of both UBL domains, while K48-Ub$_2$ is recognized mainly through the proximal Ub. These data, together with NMR and cross-linking mass spectrometry (XL-MS), suggest that PLpro$^{CoV-2}$ interacts with both UBL domains of ISG15, whereas K48-Ub$_2$ is recognized largely through the proximal Ub, with distal Ub contributing to binding through alternative dynamic interactions. We used modeling to predict alternative modes of PLpro binding to substrates that are consistent with cross-linking data. Finally, we tested our binding models by performing an in silico ΔΔG alanine scan on PLpro$^{CoV-2}$ in complex with K48-Ub$_2$/hISG15 substrates and experimentally validated their binding effects to show differential domain utilization by the PLpro for the two substrates. Our findings uncover binding heterogeneity in PLpro interactions with hISG15 and ubiquitin substrates that decouples binding affinity from the enzyme proteolytic activity.

## Results
### Sequence and topological differences between hISG15 and K48-Ub$_2$

Recent biochemical binding and cleavage assays have shown that PLpro$^{CoV-1}$ prefers K48-Ub$_2$ while the related PLpro$^{CoV-2}$ binds more tightly to both human and mouse ISG15 (hISG15 and mISG15). A nearly 20-fold higher affinity compared to K48-Ub$_2$ suggests that the sequence variation at the substrate binding interface between PLpro$^{CoV-1}$ and PLpro$^{CoV-2}$ may dictate substrate specificity[26,27]. Importantly, these previous studies on PLpro$^{CoV-1}$/PLpro$^{CoV-2}$ binding to Ub$_2$ used a non-hydrolyzable synthetic triazole linker between the Ubs rather than a native isopeptide K48 linkage, raising questions how linker geometry and rigidity may influence binding to PLpro[26–28]. When considering both domains in K48-Ub$_2$ and hISG15, they are 33% identical in sequence (Supplementary Fig. 1a), while the distal (N-terminal) UBL domain of hISG15 is 29% identical to Ub, and the proximal (C-terminal) UBL domain of hISG15 has a slightly higher sequence identity of 37%. hISG15 and mISG15 are 63% sequence identical, and both have similar sequence identities to Ub (Supplementary Fig. 1a, b). Intriguingly, however, a recent study reported that hISG15 binding to PLpro$^{CoV-2}$ has an order of magnitude higher on- and off-rates than mISG15 binding[26]. Importantly, Ub and UBLs of hISG15 and mISG15 also vary in the binding surfaces (Supplementary Fig. 1b, c). The protein domains in hISG15 and K48-Ub$_2$ have homologous folds, but their sequences and topologies of how the two domains are linked are different (Fig. 1a and Supplementary Fig. 1d).

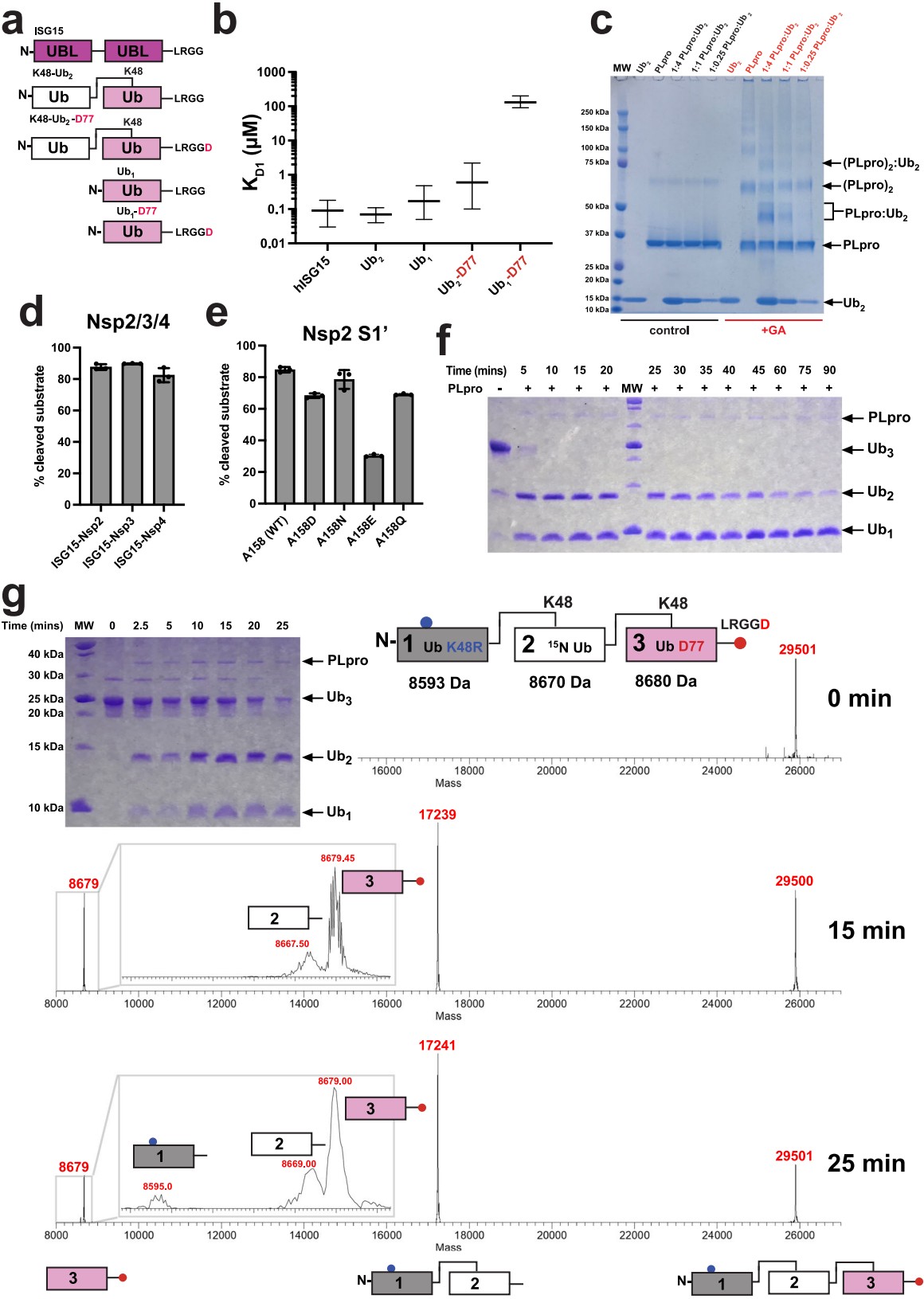

The earlier reported structure of the PLpro[CoV-1]:K48-Ub$_2$ complex shows the proximal Ub bound to the Zn finger and palm domains via its surface hydrophobic patch (comprising residues L8, I44, and V70), placing the C-terminal tail modified with allylamine in a groove and covalently linked to active site C111 (Supplementary Fig. 1d)[28]. A recent structure of full-length mISG15 bound to PLpro[CoV-2 27] revealed a distinct binding mode of the proximal and distal UBL domains of mISG15. The proximal UBL is shifted away from the finger domain compared to the proximal-Ub binding mode while still placing the C-terminal LRGG tail into the active site of the protease (Supplementary Fig. 1d). Comparison of ISG15 and Ub recognition surfaces reveals that the hydrophobic patch centered on I44 in Ub (Supplementary

**Fig. 1 | PLpro$^{CoV-2}$ substrate binding and recognition. a** Topology of utilized substrates: ISG15, K48-linked Ub$_2$ ending with G76 on the C-terminus (K48-Ub$_2$), K48-linked Ub$_2$ ending with D77 on the C-terminus (K48-Ub$_2$-D77), and corresponding monomeric ubiquitins. **b** MST analysis of substrates binding to PLpro$^{CoV-2}$. Comparison of dissociation constants for the primary binding event (K$_{d1}$). Shown mean value of K$_{d1}$ is derived from a fit to three independent experiments with error bars corresponding to a 68.3% confidence interval derived from error-surface projections. **c** Inter- and intramolecular non-specific cross-linking of PLpro$^{CoV-2}$ complexes. Glutaraldehyde cross-linked samples with different molar ratios reveal the formation of heterogeneous covalent PLpro$^{CoV-2}$:Ub$_2$ heterodimer complex bands (red) compared to untreated reactions (black) by SDS-PAGE. This experiment was repeated at least two times, yielding similar results. **d** Quantification of the cleavage efficiency of hISG15 C-terminal fusions with peptides from Nsp2 (AYTRYVDNNF), Nsp3 (APTKVTFGDD), and Nsp4 (KIVNNWLKQL) from SARS-CoV-2 that mimic natural substrates of PLpro$^{CoV-2}$, as revealed by SDS-PAGE gel. Shown is

the percentage of the input population cut by PLpro$^{CoV-2}$. Each experiment was performed in triplicate and the results are reported as an average with a standard deviation. **e** Quantification of the cleavage efficiency of Nsp2 peptides as dependent on amino acid located at the C-terminus of LRGG(X) motif. Shown is the percentage of the input population cut by PLpro$^{CoV-2}$. Each experiment was performed in triplicate and the results are reported as an average with a standard deviation. **f** PLpro$^{CoV-2}$ cleavage of K48-Ub$_3$. SDS-PAGE gel reveals that PLpro cleaves Ub$_3$ into Ub$_2$ and Ub$_1$ efficiently at various rates. **g** Mass spectrometry detection of cleavage patterns for PLpro$^{CoV-2}$ hydrolyzing Ub$_3$, in which the distal Ub (1) carries K48R mutation, the endo (2) Ub is $^{15}$N-labeled, and the proximal (3) Ub contains C-terminal D77 extension. Analysis of the time course reveals that this Ub$_3$ is primarily hydrolyzed between Ubs 2 and 3. Masses of individual Ub units are shown on the top, and the identified products are shown at the bottom. Experiments in panels (**f**, **g**) were repeated independently at least two times with similar results, mass

Fig. 1b, c) is more polar in the proximal UBL of hISG15 and mISG15. Given that the prior studies used a triazole-linked Ub dimer, we wanted to test the influence of the linker composition on binding to PLpro$^{CoV-2}$. We used microscale thermophoresis (MST) binding experiments to quantify affinity between PLpro$^{CoV-2}$ and three substrates: hISG15, K48-Ub$_2$, and Ub$_1$ (Fig. 1a). Additionally, we tested PLpro$^{CoV-2}$ binding to K48-Ub$_2$ and Ub$_1$ containing a C-terminal aspartic acid (D77) after the LRGG PLpro recognition site (Fig. 1a), which is typically used for controlled enzymatic synthesis of ubiquitin chains[29]. Fitting our binding data to a 1:1 binding model (Supplementary Fig. 2a) resulted in abnormally high $\chi^2$ and systematic deviation in the residuals (Supplementary Fig. 2b). Improved fits were observed using a model that assumes two binding events with different binding constants (Supplementary Fig. 2a) yielding statistically significant reductions in the $\chi^2$ compared to one binding event for all datasets except Ub$_1$-D77 (Supplementary Fig. 2b). We find that PLpro$^{CoV-2}$ binds both hISG15 and K48-Ub$_2$ with high affinity (90 and 70 nM, respectively) and more strongly than Ub$_1$ (apparent K$_d$, 170 nM) (Fig. 1b and Supplementary Table 1) although the actual microscopic K$_d$ of Ub$_1$ could be even lower as it may be able to bind to multiple sites on PLpro$^{CoV-2}$ (see below). Interestingly, K48-Ub$_2$ with a C-terminal aspartic acid (K48-Ub$_2$-D77) binds almost tenfold weaker (600 nM) compared to Ub$_2$, and Ub$_1$-D77 exhibited weak binding (130 μM) (Fig. 1b and Supplementary Table 1). This is consistent with a lack of reported protease substrates with acidic residues at the C-terminus of the LRGG(X) motif[6]. Our analysis also suggests the presence of secondary binding events for hISG15 and K48-Ub$_2$ with μM affinities (Supplementary Fig. 2a and Supplementary Table 1). This is supported by cross-linking data where we observed covalent adducts with a molecular weight corresponding to heterodimers (PLpro:substrate) but also heterotrimers ((PLpro)$_2$:substrate) (Fig. 1c and Supplementary Fig. 2c).

The observed heterogeneity of the species formed suggests that ISG15 may bind in a more defined orientation to PLpro while Ub$_2$ appears to bind in several arrangements. As a comparison to Ub$_1$, we also measured affinities for the isolated N-terminal (distal, hISG15$_{distal}$) and C-terminal (proximal, hISG15$_{prox}$) UBLs from hISG15 and found that they bind with micromolar dissociation constants similar to Ub$_1$-D77 (Supplementary Fig. 2d and Supplementary Table 1) and consistent with NMR measurements (Supplementary Fig. 2e). The significantly weaker PLpro binding to the isolated UBLs compared to full-length hISG15 indicates that both UBL domains are required for the high-affinity binding of hISG15. These experiments indicate that there is a dominant binding mode between the substrate and PLpro, but higher-order complexes are also possible, thus justifying the need to fit our binding data with more complex binding models. This is more pronounced for Ub$_2$ than for ISG15, as these different states appear to contribute to a higher affinity of PLpro for Ub$_2$ as compared to Ub$_1$. This may be explained by the flexibility of the Ub-Ub (isopeptide)

linker in K48-Ub$_2$ that enables this dimer to adopt heterogeneous conformational ensembles[30–34].

Derived from published structures of mISG15:PLpro$^{CoV-2}$ and Ub$_2$:PLpro$^{CoV-2}$[26–28], we anticipated that hISG15 and Ub$_2$ bind PLpro utilizing both UBL/Ub domains. However, the relatively small difference in Ub$_2$ and Ub$_1$ affinities suggests that the second Ub contributes modestly to the binding. Additionally, Ub$_1$-D77 binds nearly three orders of magnitude more weakly compared to WT Ub$_1$, while Ub$_2$-D77 yields an affinity more similar to Ub$_1$, perhaps indicative of a change in binding mode primarily utilizing a single Ub. To explore how affinity relates to PLpro proteolytic activity, we conducted PLpro$^{CoV-2}$ cleavage assays for hISG15 with modified C-terminal tails mimicking natural SARS-CoV-2 substrates or K48-linked Ub$_3$/Ub$_2$. We found that PLpro$^{CoV-2}$ can efficiently cleave peptides containing the LRGG motif. The hISG15 fusions to fragments of Nsp2, Nsp3, and Nsp4 are proteolyzed with similar rates (Fig. 1d and Supplementary Fig. 2f). We investigated how amino acid X (position 158) at the C-terminus of the LXGG↓(X) motif can impact cleavage. When Ala of Nsp2 peptide is substituted with Glu, the fusion peptide is being cut the slowest (Fig. 1e and Supplementary Fig. 2g), consistent with binding measurements for Ub$_1$-D77 and Ub$_2$-D77 which have an additional acidic amino acid on the C-terminus (Fig. 1b and Supplementary Table 1). We also found that PLpro$^{CoV-2}$ hydrolyzes K48-Ub$_3$ to Ub$_2$ and Ub$_1$ rapidly, but the subsequent cleavage of Ub$_2$ to Ub monomers is slow (Fig. 1f, g). Finally, we found that Ub$_2$-D77 is cleaved more rapidly compared to Ub$_2$, which suggests that despite Ub$_2$-D77 binding with a notably lower affinity to PLpro$^{CoV-2}$ it must be bound differently than Ub$_2$ to enable the productive hydrolysis of Ub$_2$ (Supplementary Fig. 2h). If Ub$_3$ binds predominantly with two Ub units, there are two possible binding modes of Ub$_3$ on PLpro.

To test which binding mode is dominant, we used a K48-Ub$_3$ that contains three distinct Ub units: 1 (mutant Ub-K48R, distal domain), 2 (U-$^{15}$N-labeled Ub, endo (middle) domain) and 3 (Ub-D77, proximal domain) (Fig. 1g), allowing mass spectrometry-based identification of each cleavage product. Analysis by MS of a cleavage time course of this Ub$_3$ construct reveals that the first cleavage occurs between Ub units 2 and 3, releasing the C-terminal Ub$_1$-D77 (i.e., proximal) with only minor products for the other Ubs (Fig. 1g). This is consistent with Ub units 1 and 2 bound to PLpro with the C-terminal tail of unit 2 fitting the active site for rapid hydrolysis (Fig. 1g). Three Ub binding sites (S2, S1, and S1′) were proposed for PLpro[27]. In this model, Ub unit 1 would bind to S2, unit 2 to S1, and unit 3 to S1′ (Supplementary Fig. 1d). Interestingly, Ub$_2$ could bind to PLpro in two modes. The high-affinity mode, as in Ub$_3$ binding, where Ub unit 1 (distal) binds to S2 and unit 2 to S1, results in no cleavage. Only the second, lower-affinity mode is productive for Ub$_2$ cleavage, wherein Ub unit 1 is bound to S1 and unit 2 (proximal Ub) located at S1′. In this arrangement, the C-terminal tail of unit 1 connecting the two Ubs is placed in the active site of PLpro, and Ub$_2$ is cut

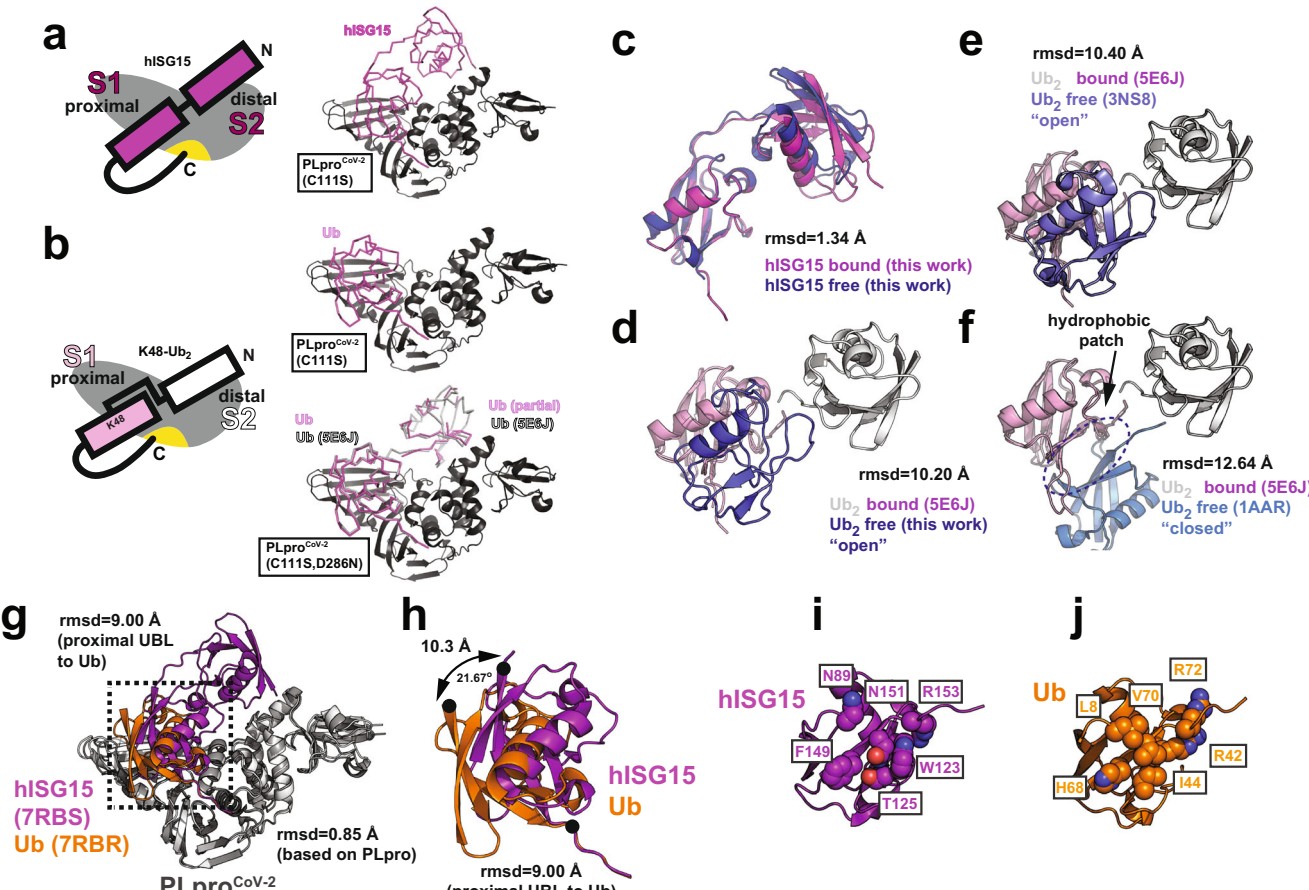

**Fig. 2 | MX structure of PLpro^CoV-2 bound to human ISG15 and K48-Ub_2 reveals differential usage of distal domains.** Schematic of PLpro bound to (**a**, left) hISG15 and (**b**, left) K48-linked Ub_2. PLpro is shown in gray, with an active site indicated in yellow. hISG15 is shown in magenta. Proximal and distal Ubs are pink and white, respectively. Crystal structures of PLpro^CoV-2 bound to (**a**, right) hISG15 and (**b**, right) K48-Ub_2. PLpro (C111S or C111S, D286N) is shown in cartoon representation, hISG15 and K48-Ub_2 are shown as backbone traces. PLpro, hISG15, and K48-Ub_2 are colored as in (**a**, left) and (**b**, left). **c** Overlay of crystal structures of bound hISG15 (magenta, PDB ID: 7RBS, this study) and unbound hISG15 (blue, PDB ID: 7S6P, this study). **d** Overlay of bound conformation of K48-linked Ub_2 observed in complex with PLpro^CoV-1 (PDB ID: 5E6J) with unbound conformation of K48-linked Ub_2 (PDB ID: 7S6O, this study). The proximal Ub of both bound and unbound conformations is shown in pink, the distal Ub of bound conformation is shown in white, and the distal Ub of unbound conformation is shown in blue. **e** Overlay of bound conformation of

K48-linked Ub_2 observed in complex with PLpro^CoV-1 (PDB ID: 5E6J) with unbound open conformation of Ub_2 (PDB ID: 3NS8). Ub units are shown as in (**d**). **f** Overlay of bound conformation of K48-linked Ub_2 observed in complex with PLpro^CoV-1 (PDB ID: 5E6J) with unbound closed conformation of Ub_2 (PDB ID: 1AAR). Ub units are shown as in (**d**). The intramolecular Ub-Ub interface is indicated with an arrow. **g** Structural overlay of PLpro^CoV-2:hISG15 and PLpro^CoV-2:Ub_2 (PDB ID: 7RBR, this study). Proteins are shown in cartoon and colored gray (PLpro), magenta (hISG15), and orange (Ub_2). **h** Zoom in of the boxed area in (**g**) representing overlay of the proximal domain of hISG15 and Ub. The proximal domain of hISG15 and Ub are represented as in (**g**). Rotation of the binding surface is indicated with arrows. **i, j** Comparison of the binding surfaces of hISG15 (**i**) and Ub_1 (**j**). The proximal domain of hISG15 and Ub are represented as in (**g**). Key residues are shown in space fill representation.

into monomers. These two modes of binding are competitive and because there is a difference in affinity, the rate of cleavage is reduced, eventually resulting in the complete disassembly of Ub_2.

To further characterize the cleavage of Ub_3 bound in different arrangements, we designed an NMR-based experiment using Ub_3 or Ub_3-D77 in which the endo Ub (unit 2) was ^15N-labeled (as in the MS-based assay), allowing us to simultaneously monitor in real time the signal intensities for the isopeptide NH group of K48 (Supplementary Fig. 3a, red) and for G76 (Supplementary Fig. 3a, blue) of the endo Ub as a proxy for cleavage of Ub_3 bound in two different geometries (Supplementary Fig. 3a, b). The signal corresponding to the conjugated C-terminal G76 of the endo Ub decreased rapidly (Supplementary Fig. 3c, blue triangles) and concomitantly with a rapid increase in its unconjugated G76 signal (Supplementary Fig. 3c, blue circles), indicating cleavage of the linker between the endo and proximal Ubs (units 2 and 3). By contrast, the K48 isopeptide signal disappeared much slower, indicating slower cleavage of the linker between the distal and endo Ub units (1 and 2), in full agreement with

the MS-based cleavage assay (Fig. 1g, also 1f). For this sequence of Ub_3 cleavage events to happen, Ub units 1, 2, and 3 must first occupy the S2, S1, and S1' sites, respectively, to enable cleavage of the proximal Ub (unit 3), whereas the binding arrangement where Ub unit 1 occupies the S1 site and unit 2 the S1' site, required for the distal-endo (1-2) linkage cleavage, occurs only after the proximal Ub (unit 3) gets cleaved.

Thus, our experiments uncover more complex alternate binding modes and sequence dependence of PLpro for two related substrates that have not been described to date. Our data also provide an alternative interpretation of recently published work[26,27] as the composition and flexibility of the Ub-Ub linker can significantly impact the binding or cleavage or both for Ub_2 and other protein substrates. This may explain the previously observed difference in affinity of PLpro^CoV-2 between Ub_2 and ISG15 which seems likely attributed to changes in mode of binding and/or conformational flexibility of the substrate linkage rather than mutations in the PLpro enzyme. Again, this is consistent with observed conformational flexibility of free K48-Ub_2[33,34]

and much lower conformational diversity of ISG15 as a free protein and in the complex with PLpro and other proteins (complex of hISG15 with the NS1 protein of influenza B virus[35]) (see below).

## Dual vs single domain substrate recognition determines PLpro[CoV-2] selectivity

We further investigated details of the interaction between the PLpro[CoV-2], hISG15, and K48-Ub$_2$ to reveal similarities and differences. We determined crystal structures of PLpro[CoV-2] with an active site C111S mutation, which inactivates PLpro, in complex with hISG15 at 2.98 Å resolution (Fig. 2a and Supplementary Table 2) and with K48-Ub$_2$ at 1.88 Å resolution (Fig. 2b, Supplementary Fig 4a, and Supplementary Table 2). For the PLpro[CoV-2]:hISG15 complex, we observe well-resolved electron density for the proximal and distal UBL domains bound to S1 and S2 sites, respectively (Fig. 2a). By contrast, for the PLpro[CoV-2]:K48-Ub$_2$ structures determined at much higher resolution we observe strong electron density for the proximal Ub bound to S1 site with only weak signal for the distal Ub in S2 site and no electron density for Ub in S1' site (Fig. 2b and Supplementary Fig. 4b). Despite weak electron density for the distal Ub, some regions of a density map resemble α-helix and amino acid chains corresponding to portions of the distal Ub from the superposed structure of PLpro[CoV-1] bound to K48-Ub$_2$ (PDB ID: 5E6J) upon only slight adjustment (Supplementary Fig. 4c). Therefore, there is sufficient room in our crystals of PLpro[CoV-2]-C111S:K48-Ub$_2$ to fit the distal Ub at site S2, but the binding mode of the distal Ub may be less defined (Supplementary Fig. 4d)[28]. An alternative explanation is that PLpro[CoV-2] (C111S) can hydrolyze the K48 linkage slowly, leading to a mixture of Ub$_1$ and Ub$_2$. To test this directly, we ran SDS-PAGE gels of our crystals and observed predominantly Ub$_2$ with only a minor Ub$_1$ species (Supplementary Fig. 4e).

We have also determined the structure of the PLpro[CoV-2]-C111S, D286N double-mutant:K48-Ub$_2$ complex(Supplementary Fig. 4f and Supplementary Table 2). These crystals diffracted to higher resolution (1.45 Å) and allowed to model more fragments of distal Ub when compared to the 1.88 Å structure of PLpro[CoV-2]-C111S:K48-Ub$_2$ (PDB ID: 7RBR) (Supplementary Fig. 4f, g). In total, 41 of 76 distal-Ub residues were built using this electron density map. Occupancies for most residues were lower and determined based on R-factor values and inspection of $F_o - F_c$ difference maps. The location of the distal Ub fragment closely matches the position of distal Ub in the structure of PLpro[CoV-1] bound to a K48-Ub$_2$ (PDB ID: 5E6J). It is important to mention that the insertion of complete distal Ub causes clashes with neighboring molecules, suggesting that multiple conformations must be present in the crystal of PLpro[CoV-2]-C111S, D286N:K48-Ub$_2$. Our structures revealed how the protease differentially recognizes hISG15 using both UBL domains, while K48-Ub$_2$ is predominantly recognized using the proximal Ub.

We additionally determined structures of the free hISG15 and K48-Ub$_2$ to 2.15 and 1.25 Å resolution, respectively. The bound and free hISG15 conformations are similar with a root-mean-square deviation (rmsd) for Cα atoms of 1.34 Å (Fig. 2c) based on the alignment of the C-terminal (proximal) UBL. An analogous comparison of our unbound K48-Ub$_2$ structure with the only known PLpro-bound conformation of K48-Ub$_2$ (from SARS-CoV-1)[28] revealed an rmsd of 10.2 Å. The Ub units are oriented very differently relative to each other (Fig. 2d), with the PLpro-bound Ub$_2$ in an extended conformation (like hISG15) while the unbound Ub$_2$ in an open conformation with the functional hydrophobic surface patches exposed for binding. We also compared the bound conformation with two other published canonical open (Fig. 2e) and closed (Fig. 2f) conformations of K48-Ub$_2$, revealing large differences in rmsd of 10.4 Å (PDB ID: 3NS8) and 12.64 Å (PDB ID: 1AAR), respectively. In the open conformation, the functional binding surface on Ub is exposed, while in the closed conformation, the functional nonpolar surfaces are engaged in intramolecular Ub-Ub interactions (Fig. 2f, arrow). Interestingly, our K48-Ub$_2$ structure is nearly identical

to the previously reported 'open' conformation (PDB ID: 3NS8) with an rmsd of 0.23 Å[30]. These data reflect that hISG15 is more rigid while the K48-Ub$_2$ exists as an ensemble of conformational states, which may influence binding and recognition by PLpro and other USPs[31–33]. Moreover, different Ub linkage types may exploit different interdomain conformational spaces explaining the source of specificity[34]. These observations confirm that PLpro is capable of recognizing distinct surfaces presented on Ub or UBL dimers (see below).

## Functional surfaces of hISG15 and K48-Ub$_2$ are recognized differentially by PLpro[CoV-2]

We first compared the binding modes of hISG15 and proximal Ub in our structures. The structures of PLpro in both complexes are similar with an rmsd of 0.85 Å, but the binding surface contacts of the proximal Ub are shifted towards the fingers domain of PLpro compared to the proximal UBL domain of ISG15 (Fig. 2g, h). This shift in the binding mode is manifested by a 21.7° rotation around the C-termini of hISG15 and Ub, displacing the N-terminal residue by 10.3 Å (Fig. 2h). This is despite the structural homology between the hISG15 proximal UBL domain and Ub (rmsd of 0.96 Å) (Supplementary Fig. 5a); thus it is likely dictated by differences in PLpro binding to the Ub and UBL interacting surfaces (Supplementary Fig. 5b).

We also compared our structure to the previously published PLpro[CoV-2]:mISG15[26] complex. Overlay of the two structures (PDB IDs: 7RBS and 6YVA) reveals the good structural similarity with the overall rmsd of 0.70 Å for PLpro and 1.40 Å for hISG15 and mISG15 (Supplementary Fig. 6a). The proximal UBL domains of both ISG15s are well aligned and make several conserved interactions with the PLpro but interaction with the distal UBL domain shows the largest deviation (Supplementary Fig. 6a). We compared the contacts from the distal domain of mISG15 and hISG15 to previously determined hotspot residues (F69 and V66) on PLpro[CoV-2][26]. We find that in the PLpro[CoV-2]:mISG15 structure K30 and M23 of the distal UBL of mISG15 interact with F69 of PLpro[CoV-2], while V66 of PLpro[CoV-2] interacts with A2 of the substrate (Supplementary Fig. 6b). By contrast, in our PLpro[CoV-2]:hISG15 structure residue 30 of hISG15 is an alanine, thus leaving M23 alone to stabilize the interaction with F69 of PLpro[CoV-2], while the N-terminus of hISG15 interacts with V66 of PLpro[CoV-2] (Supplementary Fig. 6c). Residue 20 in ISG15 makes similar nonpolar contacts with V66 but it varies between the mouse (T20) and human (S20) protein (Supplementary Fig. 6b, c). We additionally compared the interactions between the proximal UBL domains of the mISG15 and hISG15 where the UBL binds in a similar binding mode (Supplementary Fig. 6d). We find that overall, the two ISG15 proteins make similar, but not identical contacts determined by the sequence variation between mISG15 and hISG15. This suggests that the virus may have a different impact if it infects distinctive species. The central interacting residues on PLpro[CoV-2] are Y171, E167, and M208, which interact with conserved R153/R151, W123/W121, and P130/P128 on the proximal domains of hISG15 and mISG15, respectively (Supplementary Fig. 6e, f). The interaction is centered on a salt bridge between E167 of PLpro[CoV-2] and R153 of hISG15, while the equivalent arginine (R151) in mISG15 is not oriented properly to form a salt bridge. Nonetheless, this core interaction is stabilized by nonpolar interactions of the surrounding residues from both sides of the interface, including Y171 of PLpro[CoV-2] and W123/W121 and P130/P128 from hISG15 and mISG15, respectively (Supplementary Fig. 6e, f). By contrast, interactions with R166 of PLpro[CoV-2] vary more significantly between mISG15 and hISG15. In the mISG15 structure, the side chain of M208 is not resolved, while in the hISG15 structure, M208 packs against R166 (Supplementary Fig. 6f). Interestingly, R166 forms a salt bridge with E87 of mISG15 which is changed to asparagine (N89) in hISG15 (Supplementary Fig. 6e, f). To compensate for this loss of interaction, N151 of hISG15 makes a hydrogen bond with R166 (Supplementary Fig. 6e, f). This highlights subtle sequence changes between mISG15 and hISG15 that allow

interface rearrangements while preserving the binding mode and may explain previously reported differences in binding between human and mouse ISG15[26]. Our structures show that hISG15 binds PLpro$^{CoV-2}$ utilizing both proximal and distal UBL domains (Fig. 2a), while binding of K48-Ub$_2$ is primarily driven by interaction with the proximal Ub with only weak density observed for the distal Ub (Fig. 2b and Supplementary Fig. 4b). We also compared our PLpro$^{CoV-2}$:hISG15 structure to a recent structure of PLpro$^{CoV-2}$ bound to only the proximal domain of hISG15[27] (Supplementary Fig. 7a; PDB ID: 6XA9, 2.9 Å resolution). As in the PLpro$^{CoV-2}$:mISG15 complex, the structural similarity is high, with an overall Cα rmsd of 1.0 Å. A comparison of the interface contacts reveals nearly identical interactions, even preserving sidechain rotamers between the proximal hISG15 and PLpro in the two structures (Supplementary Fig. 7b). Finally, our structure of PLpro$^{CoV-2}$:K48-Ub$_2$ is nearly identical in binding mode to the previously published structure of PLpro$^{CoV-2}$:Ub$_1$ (Supplementary Fig. 7c; PDB ID: 6XAA, 2.7 Å resolution) with a Cα rmsd of 0.32 Å and nearly identical side chain rotamers at the interface (Supplementary Fig. 7d). Interestingly, our structure was determined to higher resolution and without the introduction of a covalent linkage of Ub$_2$ to PLpro$^{CoV-2}$ suggesting that the covalent linkage does not alter the physiological binding of the substrate. By contrast, however, the introduction of a synthetic Ub-Ub linker in Ub$_2$ does influence the binding of the substrate to PLpro$^{CoV-2}$.

However, PLpro cuts K48-Ub$_2$ slowly, because for productive cleavage, the distal Ub must bind to S1 (low-affinity binding) and proximal Ub occupy the S1' site. S1' site is likely less specific as it must accept multiple protein sequences. This is supported by our observations discussed earlier that K48-Ub$_3$ is cut efficiently to Ub$_2$ and Ub$_1$, with the remaining Ub$_2$ bound in a non-cleavable binding mode.

## Interactions of K48-Ub$_2$ and ISG15 with PLpro$^{CoV-2}$ in solution characterized using NMR

We then used NMR to further characterize PLpro$^{CoV-2}$ binding to hISG15 and K48-Ub$_2$ and to examine if the contacts observed in crystals also occur in solution. The addition of unlabeled PLpro$^{CoV-2}$-C111S caused substantial perturbations in the NMR spectra of $^{15}$N-labeled hISG15 (Fig. 3a). We observed the disappearance of signals of free hISG15 and the emergence of new ones; this indicates slow-exchange binding regime[36,37], consistent with the sub-μM $K_d$ values measured by MST (Fig. 1b and Supplementary Table 1). The strongly attenuated signals of hISG15 residues, including the C-terminal G157, are consistent with our crystal structure of the PLpro$^{CoV-2}$:hISG15 complex (Fig. 3a, h). A similar behavior was observed for $^{15}$N-labeled K48-Ub$_2$ upon the addition of PLpro$^{CoV-2}$ (Fig. 3b, c), where both the distal and proximal Ubs exhibited strong attenuation or disappearance of NMR signals and the emergence of new signals, primarily for residues in and around the hydrophobic patch as well as the C-termini. The affected residues mapped to the binding interface in our PLpro$^{CoV-2}$:K48-Ub$_2$ crystal structure (Fig. 3i), and the slow-exchange behavior is also consistent with the sub-μM $K_d$ values (Fig. 1b and Supplementary Table 1). The slow-exchange binding regime for both hISG15 and K48-Ub$_2$ is also generally consistent with the reported slow off-rates (0.2, 0.4 s$^{-1}$)[26]. PLpro$^{CoV-2}$ also caused noticeable perturbations in the NMR spectra of Ub$_1$, although these were weaker than in Ub$_2$, and several residues showed gradual signal shifts indicative of fast exchange, consistent with our MST measurements (Figs. 3d, 1b and Supplementary Table 1). We additionally compared binding between PLpro$^{CoV-2}$ and $^{15}$N-labeled hISG15$_{distal}$ and hISG15$_{prox}$ and detected few changes in the spectra of hISG15$_{distal}$ indicative of weak interactions (Supplementary Fig. 2e). Interestingly, titration of PLpro$^{CoV-2}$ into $^{15}$N-labeled hISG15$_{prox}$ yielded similar signal shifts and attenuations to those observed in full-length hISG15, suggesting similar binding interactions, although unbound signals were still observed even at 2x molar access of PLpro, consistent with weaker affinity. Notably, the signal attenuation of C-terminal G157

and the characteristic shifts of PLpro$^{CoV-2}$ Trp signals indicate that the C-terminus of hISG15$_{prox}$ alone can still bind to the active site of PLpro$^{CoV-2}$ (Supplementary Fig. 2e), consistent with a slightly higher affinity of PLpro$^{CoV-2}$ for hISG15$_{prox}$ compared to hISG15$_{distal}$ (Supplementary Fig. 2d).

We also performed reverse-titration NMR experiments where unlabeled hISG15, K48-Ub$_2$, or Ub$_1$ was added to $^{15}$N-labeled PLpro$^{CoV-2}$. Both hISG15 and K48-Ub$_2$ caused substantial perturbations in the $^{15}$N-PLpro$^{CoV-2}$ spectra (Fig. 3e, f). Particularly noticeable was the change in the indole NH signals of W93 and W106 located in close proximity to the active site of PLpro, as well as of imidazole NH signal attributed to the active site H272 (Fig. 3e, f, Supplementary Fig. 8a, b), in agreement with the C-termini of hISG15 and Ub$_2$ entering the active site of PLpro in our crystal structures. The addition of Ub$_1$ caused significantly lesser overall $^{15}$N-PLpro$^{CoV-2}$ signal perturbations, although the W$_ε$ and H$_δ$ signal shifts were clearly visible when Ub$_1$ was in significant excess (Fig. 3g and Supplementary Fig. 8c–e). Even at 8-molar excess of Ub$_1$, both free and bound W$_ε$/H$_δ$ signals were present, consistent with weaker binding. Taken together, the NMR data qualitatively suggest that the apparent strength of PLpro$^{CoV-2}$ binding is: hISG15 ≈ K48-Ub$_2$ > Ub$_1$, consistent with our MST data.

These NMR data indicate that binding to PLpro involves both UBLs of hISG15 and both Ubs of K48-Ub$_2$. Interestingly, despite being identical and having very similar chemical shifts in the unbound state (Supplementary Fig. 9a), the distal and proximal Ubs show markedly different signal perturbations indicative of distinct contacts with PLpro (Fig. 3b, c and Supplementary Fig. 9b). While the perturbed residues in the proximal Ub agree well with our PLpro$^{CoV-2}$: K48-Ub$_2$ crystal structure, where this Ub occupies the S1 site, several perturbations observed in the distal Ub (most notably for N25, K27, K29, D32, and K33 in the α-helix) are not observed in the PLpro$^{CoV-2}$: K48-Ub$_2$ or PLpro$^{CoV-1}$:K48-Ub$_2$ crystal structures. Thus, we cannot exclude possible additional modes of interaction between the distal Ub and PLpro$^{CoV-2}$. This might explain the low electron density for the distal Ub in the PLpro$^{CoV-2}$:K48-Ub$_2$ crystal structure. Interestingly, for several residues in Ub$_1$ the shifted signals upon addition of PLpro$^{CoV-2}$ appear at positions intermediate between those in the distal and proximal Ubs of K48-Ub$_2$ (Supplementary Fig. 9b), suggesting that Ub$_1$ might be sampling both S1 and S2 sites on PLpro.

In agreement with the MST results, our NMR data demonstrate that placement of aspartate at the C-terminus of hISG15, K48-Ub$_2$, and Ub$_1$ reduced substantially their affinity for PLpro$^{CoV-2}$, as evidenced from noticeably weaker NMR signal perturbations observed in both the D-extended substrates and PLpro (Supplementary Fig. 10). It should be mentioned that in all the NMR studies presented here the addition of PLpro$^{CoV-2}$ resulted in the overall NMR signal broadening/ attenuation reflecting an increase in the size (hence slower molecular tumbling) upon complexation with a ~36 kDa protein. The finding that hISG15 and K48-Ub$_2$ bind to the same sites on PLpro$^{CoV-2}$ enabled us to directly compare their affinities for PLpro$^{CoV-2}$ in a competition assay where hISG15 was added to a preformed PLpro$^{CoV-2}$:K48-Ub$_2$ complex, and the bound state of Ub$_2$ was monitored by $^{1}$H-$^{15}$N NMR signals of $^{15}$N-labeled proximal Ub (Fig. 3j). Titration of unlabeled hISG15 into a 1:1.5 mixture of K48-Ub$_2$ and PLpro$^{CoV-2}$ resulted in the gradual disappearance of PLpro-bound signals of Ub and the concomitant emergence of free K48-Ub$_2$ signals at their unbound positions in the spectra (Fig. 3j and Supplementary Fig. 9c). The observed decrease in the intensity of the bound signals agrees with the prediction based on the $K_{d1}$ values for K48-Ub$_2$ and hISG15 derived from our MST experiments (Fig. 3j) but not with the $K_d$ values reported previously[26] (Supplementary Fig. 9d).

Since hISG15 binds to the same PLpro$^{CoV-2}$ surface as Ub$_2$ but contains an uncleavable linkage between UBLs, we then examined if hISG15 can inhibit polyUb cleavage by PLpro. When hISG15 was added to a cleavage reaction of Ub$_3$ or Ub$_2$ by PLpro$^{CoV-2}$ it did not interfere

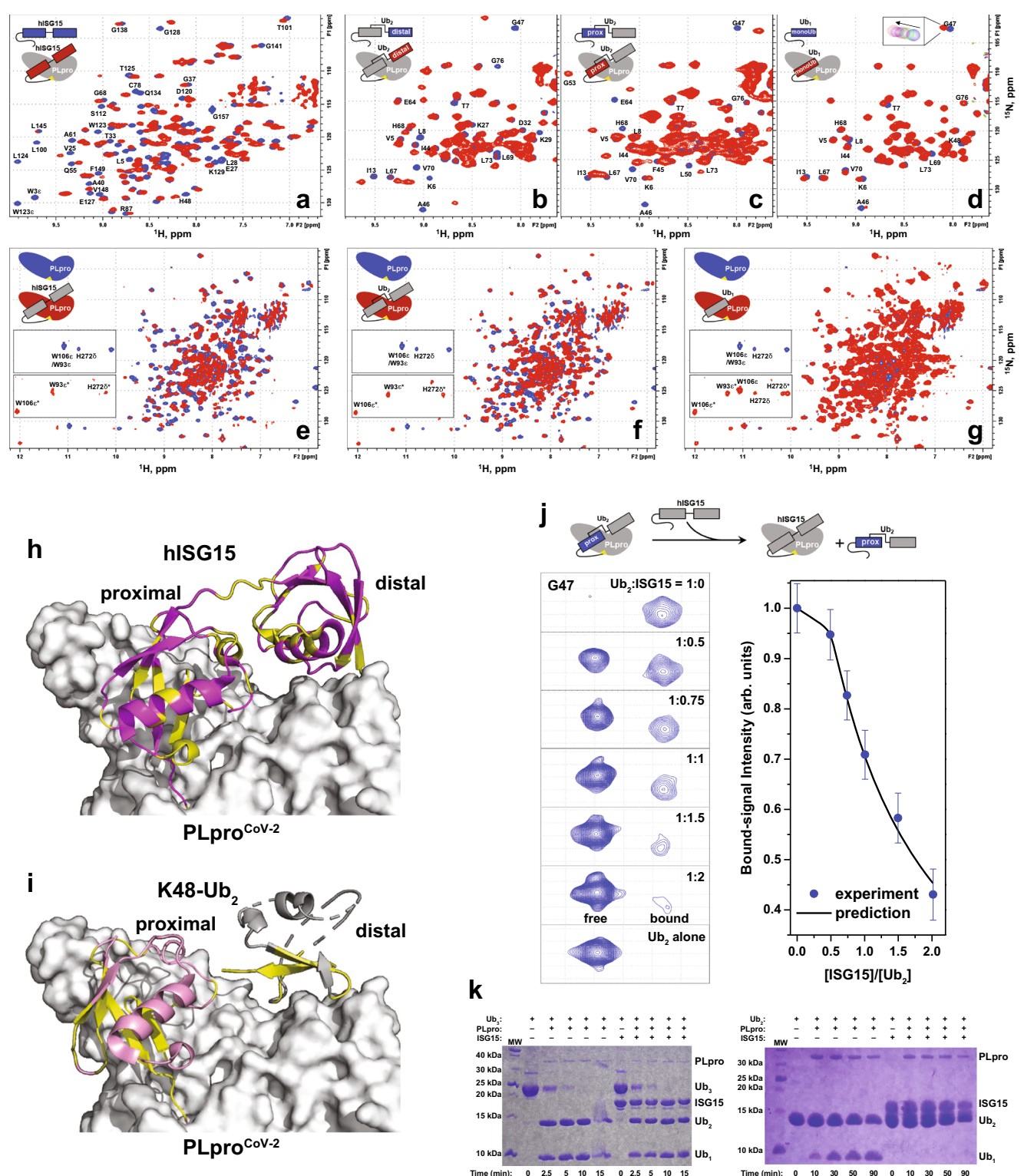

with Ub₃ cleavage to Ub₂ (Fig. 3k, left gel), but it blocked hydrolysis of Ub₂ to monomers (Fig. 3k, right gel, also Supplementary Fig. 9e). A similar effect was observed on the cleavage of Ub₂ in the presence of Ub₁ (Supplementary Fig. 9e). This can be explained by different options for productive cleavage of Ub₃ and Ub₂. The productive cleavage of Ub₃ is predominantly accomplished by binding two Ub units (2 and 1) to S1 and S2 sites on PLpro$^{CoV-2}$, respectively, and the unit 3 of Ub₃ occupying the S1′ site, thus placing the isopeptide bond on its K48 in the active site of PLpro$^{CoV-2}$. This is a high-affinity Ub₃ binding,

and hISG15 and particularly Ub₁ cannot easily compete for binding. As discussed earlier, K48-Ub₂ can bind in two different modes, one with two Ub domains binding to S1 and S2 sites on PLpro, but this binding cannot result in the cleavage. In order to break the isopeptide bond between two Ubs, the distal Ub must bind to the S1 site on PLpro such that the proximal Ub will then occupy the S1′ site. The LRGG motif can then be recognized, and the isopeptide bond is cleaved. But Ub₂ binding through a single Ub unit to S1 site is of low affinity; thus, both hISG15 and Ub₁ can compete with Ub₂ and inhibit its cleavage.

**Fig. 3 | NMR data showing PLpro^CoV-2 interactions with hISG15, K48-Ub₂, and monomeric Ub. a–d** Overlay of ¹H-¹⁵N NMR spectra of ¹⁵N-labeled **a** hISG15, **b** distal Ub in Ub₂, **c** proximal Ub in Ub₂, or **d** Ub₁, alone (blue) and with 1–1.25 (2 for Ub₁) molar equivalents of unlabeled PLpro^CoV-2-C111S (red). Signals of select residues are indicated. Inset in (**d**) illustrates the gradual shift of the G47 signal during titration. **e–g** Overlay of ¹H-¹⁵N NMR spectra of ¹⁵N-PLpro^CoV-2-C111S, Y171H alone (blue) and with 1.25–1.5 molar equivalents of unlabeled **e** hISG15, **f** Ub₂, or **g** eight molar equivalents of Ub₁ (red). Insets zoom on the region containing indole HN signals of tryptophans (W93 and W106) and HN of the imidazole ring of histidine H272; the bound signals are marked with an asterisk. **h** hISG15 residues with strong signal perturbations mapped (yellow) on our structure of PLpro^CoV-2:hISG15 complex. **i** Residues in the proximal and distal Ubs with strong signal perturbations mapped (yellow) on our structure of PLpro^CoV-2:K48-Ub₂ complex. **j** Competition for PLpro^CoV-2 binding between Ub₂ and hISG15. (Left) Representative ¹H-¹⁵N NMR signals of G47 in ¹⁵N-labeled proximal Ub of Ub₂ pre-mixed with unlabeled PLpro^CoV-2-

C111S,Y171H (in 1:1.5 molar ratio) upon addition of unlabeled hISG15, for indicated values of hISG15:Ub₂ molar ratio. (Right) The intensity of the PLpro-bound signal of G47 as a function of [hISG15]:[Ub₂] (dots) and the predicted molar fraction of bound Ub₂ (line) based on the $K_{d1}$ values obtained in this work (Supplementary Table 1). The symbols depict normalized peak intensities extracted directly from the respective 2D NMR spectra and the error bars represent experimental uncertainties in intensities obtained by error propagation using the experimental noise measured over at least five different regions in the spectrum that do not contain protein signals. **k** SDS-PAGE gels showing the inhibitory effect of hISG15 on disassembly of K48-linked Ub₂ by PLpro^CoV-2 (right) and minimal effect (if any) of hISG15 on disassembly of K48-linked Ub₃ (left). The hISG15 and Ub₂ constructs used here all had G (G157 or G76) as the C-terminal residue. Cleavage assays were repeated at least two times with similar results. Source data are provided as a Source Data file.

## Specific contacts in PLpro:substrate complexes detected with XL-MS

Our structural experiments indicate differences in how hISG15 and Ub₂ are recognized by PLpro^CoV-2. To gain more insight into the proposed dynamics of the interactions, we employed an XL-MS approach (Fig. 4a). We found that the 4-(4,6 dimethoxy-1,3,5-triazin-2-yl)−4-methyl-morpholinium chloride (DMTMM) cross-linker produced robust heterodimers of PLpro^CoV-2 with hISG15, K48-Ub₂, Ub₁, or K48-Ub₂-D77 (Fig. 4b and Supplementary Fig. 11a). We identified two contacts between D61 and D62 on PLpro^CoV-2 to K35 on the distal UBL of hISG15 (Fig. 4c, 19 and 43 contacts) which map well onto our structure with the distances between carboxylates (D61 and D62) and N_ζ (K35) of 14.9 and 8.6 Å, respectively (Fig. 4c) with Cβ-Cβ distances below 30 Å consistent with the cross-linker geometry[38]. By contrast, we detected 19 cross-links between PLpro and K48-Ub₂; however, due to the sequence degeneracy between the two Ubs, we interpreted the data based on the shortest distance (Supplementary Fig. 11b). Using this strategy, 12 of the 19 observed contacts fall below a 30 Å threshold (Fig. 4d). Of these 12 contacts, 7 involve K6 from the distal Ub to the N-terminal thumb domain of PLpro^CoV-2, including E70, consistent with the distal Ub binding mode seen in the PLpro^CoV-1:Ub₂ structure (PDB ID: 5E6J)[28]. Additionally, of the 12 contacts, two between K190 on the fingers domain of PLpro^CoV-2 and E64 and E18 of Ub have 16 and 28.6 Å Cβ-Cβ distances which are compatible with the placement of the proximal Ub in the S1 binding site (Fig. 4d). Similarly to what NMR indicated, for Ub₁ we found 23 cross-links that localize to both S1 and S2 sites (Supplementary Fig. 11c). To find alternate binding sites that explain 7 of 19 identified cross-links from the PLpro^CoV-2:K48-Ub₂ dataset, we modeled 5000 binding modes of the distal Ub by docking a Ub monomer to PLpro:Ub_prox complex containing (proximal) Ub in the S1 binding site and utilizing constraints that place the docked (distal) Ub with its C-terminus in proximity to K48 of the proximal Ub (Fig. 4e, g, blue spheres). We compared the energy of the complex as a function of the sum of distances between 7 cross-linked atoms pairs that were unexplained in the initial model (Supplementary Fig. 11d). A low-energy model that explains 4 of the 7 cross-links (Fig. 4f) localizes the Ub to the PLpro's UBL and thumb domains, a site that we named S2' (Fig. 4g). In a parallel docking approach, we assumed an alternative binding mode where the distal Ub was placed in the S1 binding site (Supplementary Fig. 11e), and applied a constraint from the C-terminus of that distal Ub to K48 of the docked (proximal) Ub. Comparison of the energetics and sum of cross-linked distances, uncovered a low-energy model with the docked Ub near the PLpro's UBL that similarly explains 4 of 7 contacts (Supplementary Fig. 11e, f).

To resolve this discrepancy in binding modes, we performed XL-MS on PLpro^CoV-2:Ub₂, in which only the proximal Ub was uniformly ¹⁵N-labeled. Analysis of cross-links containing the unlabeled (distal) Ub uncovered 14 cross-links (Supplementary Fig. 11g) of which 8 are compatible with placement of this Ub in the S2 site and 7 of these again

involve K6 interacting with the N-terminal thumb domain similar to the data collected with unlabeled Ub₂ (Fig. 4d). This allowed us to propose that K48-Ub₂ can bind to PLpro in two different binding modes (Fig. 4e and Supplementary Fig. 11e). Having more confidence that the distal Ub is bound in the S2 site, we again mapped the remaining six unexplained contacts from proximally ¹⁵N-labeled dataset onto the docked model in which we sampled movement of the distal Ub (Fig. 4e); this model can explain 4 additional contacts (Supplementary Fig. 11h). Finally, we also interpreted XL-MS data for PLpro^CoV-2:Ub₂-D77 and using a model derived from the sampling of the S1' site (Supplementary Fig. 11e). We can explain the two contacts consistent with alternate binding modes of Ub₂-D77 (Supplementary Fig. 11i) with one contact comprising the distal Ub bound to the S1 site and the second involving the proximal Ub bound to the UBL domain of PLpro (S1' site) to produce a cleavage-competent binding mode. This change in binding mode may explain faster cleavage kinetics of Ub₂-D77 compared to Ub₂ (Supplementary Fig. 2h). Our combined experiments not only reaffirm the dominant binding modes between PLpro and hISG15 and K48-Ub₂ (to S1 and S2 sites) but also begin to clarify alternate binding modes that explain the heterogeneity of binding to S1 and S1' sites.

## Identification of specificity-determining sites in PLpro^CoV-2:Ub₂ and PLpro^CoV-2:hISG15 complexes

To better understand the energetic contribution of the residues at the PLpro:substrate interfaces, we applied an in silico alanine scan approach[39] to PLpro^CoV-2 in complexes with hISG15 and K48-Ub₂ (Fig. 5a). We first identified PLpro interface residues that contact the substrates and employed Rosetta[39] to calculate $\Delta\Delta G_{binding}$ comparing WT and alanine mutants. Analysis of $\Delta\Delta G_{binding}$ for PLpro^CoV-2 with the two substrates revealed interaction hotspots, most notably PLpro^CoV-2 residues E167/R166/Y264 and F69, for stabilizing S1 and S2 Ub/UBL binding sites in K48-Ub₂ and hISG15 substrates (Fig. 5b, c). Additionally, we find that for PLpro^CoV-2, the S2 site has a preference for hISG15 (Fig. 5b, c, colored in red), but the S1 site has an overall preference for Ub (Fig. 5b, c, colored in blue). We additionally interpreted the interface energetics for models of PLpro^CoV-2 in complex with Ub₂ using two geometries of the distal Ub derived from the PLpro^CoV-1:Ub₂ structure and our PLpro^CoV-2-C111S,D286N:Ub₂ structure with a partial distal Ub (Supplementary Fig. 12, described in Methods). For K48-Ub₂, the primary interaction is with the proximal Ub domain in the S1 site, but additional interaction comes also from the distal Ub interacting with the S2 site contributing to the stronger binding. In protein complexes, residues that surround protein interaction hotspots typically play important roles in determining specificity[40]. Indeed, in the S1 site, Y171 provides more stabilization for hISG15 compared to Ub₂ (Fig. 5b). These analyses highlight how the prediction of binding energetics combined with structural data can help interpret the dynamics of domain binding.

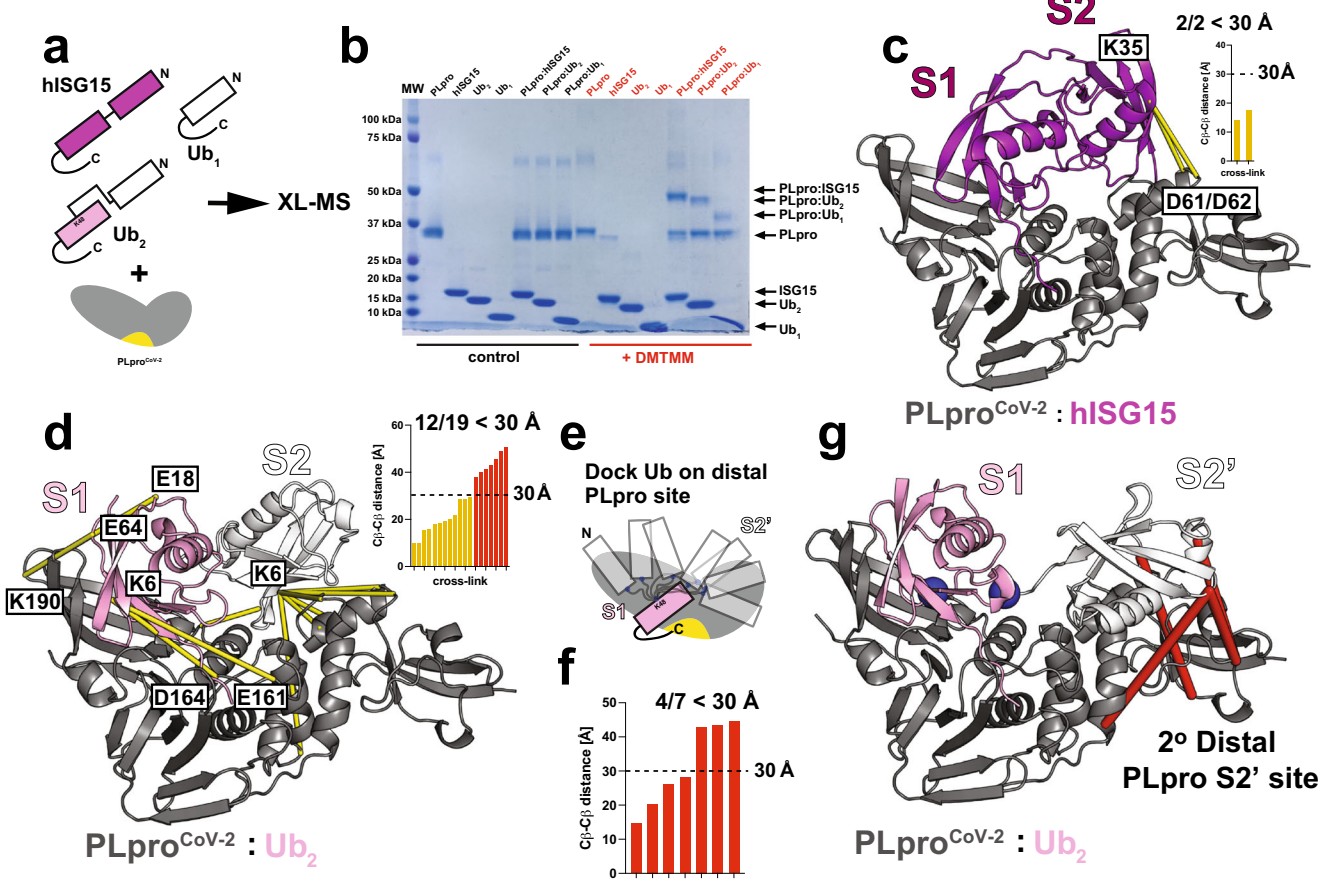

**Fig. 4 | Cross-linking mass spectrometry (XL-MS) analysis of PLpro in complex with hISG15 and K48-Ub₂. a** Schematic illustration of cross-linking mass spectrometry experiments for heterodimer complexes of PLpro^CoV-2^ with hISG15, K48-Ub₂, and Ub₁. The active site of PLpro^CoV-2^ is indicated in yellow. **b** Cross-linked samples reveal the formation of covalent heterodimer complex bands (red, DMTMM cross-linker) compared to untreated reactions (black, control) by SDS-PAGE. This experiment was repeated three times with similar results. **c** Interactions between PLpro^CoV-2^ (D61/D62) and the distal hISG15 domain (K35) identified by XL-MS mapped on the structure of heterocomplex. Both identified contacts are shorter than 30 Å. PLpro (gray) and hISG15 (magenta) are shown in cartoon representation. Cross-links are colored yellow. **d** Interactions between PLpro^CoV-2^ and K48-Ub₂ identified by XL-MS mapped on the structure of the heterocomplex. Twelve contacts found to be shorter than 30 Å show interaction of K48-Ub₂ with fingers, palm,

and thumb domains of PLpro^CoV-2^. PLpro^CoV-2^ is shown in cartoon representation and colored gray. K48-Ub₂ is shown in cartoon representation and colored white and pink for the proximal and distal Ubs, respectively. Cross-links are colored yellow. **e** Modeling strategy showing docked Ub monomer to PLpro^CoV-2^:Ub complex with Ub placed into the S1 binding site. Utilized constraint maintaining proximity between K48 of the proximal Ub and C-terminus of the docked Ub is shown in blue spheres. **f** A low-energy model generated with the strategy presented in (**e**) explains four of the remaining seven contacts from the XL-MS in (**d**). **g** Contacts shorter than 30 Å between low-energy model of PLpro^CoV-2^ bound to Ub₂ in the S1 and S2′ sites as shown in (**e**, **f**). Cross-links between distal Ub and thumb and UBL domains of PLpro^CoV-2^ are colored red. Constraint used in docking (**e**) is indicated as blue spheres. PLpro^CoV-2^ and K48-Ub₂ are shown as in (**d**). Source data are provided as a Source Data file.

Our in silico analysis of the PLpro:substrate interfaces predicts hotspot sites that are important for PLpro^CoV-2^ binding hISG15 and Ub₂ but also sites that may discriminate binding between these two substrates. Guided by these predictions, we tested four mutants, two in the S2 site (F69A and E70A) and two in the S1 site (R166A and E167A), that have different predicted effects on binding to Ub₂ and hISG15 (Fig. 5b, c). We again used MST to measure binding affinities for these alanine mutants to test their effect on PLpro^CoV-2^ binding to hISG15 and Ub₂. We evaluated both 1:1 and 1:2 binding models determining that the 1:1 fits were sufficient to explain the binding profiles (Supplementary Fig. 13a, b and Supplementary Table 1). Consistent with the computational predictions, PLpro^CoV-2^-E167A has a dramatically reduced affinity for hISG15, Ub₂, and Ub₁ (Fig. 5f–h, blue curve and Fig. 5i, blue bars). This interaction formed between E167 and R153 of hISG15 or R42 of Ub is important and stabilizing (Supplementary Fig. 13c) and is observed in the crystal structures. By contrast, our predictions suggested that R166A should have a more moderate effect on binding to the substrate as R166 forms hydrogen bond with similar geometries to N151 or Q49 of hISG15 and Ub₂, respectively (Supplementary Fig. 13c).

In the MST experiment, R166A has a larger effect on hISG15 binding compared to Ub₂ or Ub₁ (Fig. 5f–h, green curve and Fig. 5i, green bars). Our calculations of interface energetics also indicated that PLpro^CoV-2^-F69A should have a larger effect on interactions with hISG15 compared to Ub₂, including the orientation observed in the distal Ub from the partial model of our structure (Supplementary Fig. 13). Indeed, we find that PLpro^CoV-2^-F69A alters binding to hISG15 significantly and only has a modest effect on Ub₂ or Ub₁ binding (Fig. 5f–h, orange curve and Fig. 5i, orange bars). A closer look at the F69 interacting residues reveals that hISG15 uses M23 to pack against the phenylalanine, while the shorter side chain of I44 in Ub₂ is more distant. Finally, we evaluated PLpro^CoV-2^-E70A binding to hISG15, Ub₂, and Ub₁. Our predictions suggested that E70A should only weakly decrease the stability of PLpro^CoV-2^:hISG15 and PLpro^CoV-2^:Ub₂ complexes (Fig. 5b and Supplementary Fig. 12), but the MST measurements show that PLpro^CoV-2^-E70A cannot bind well to Ub₂ or Ub₁ while binding to hISG15 is less affected (Fig. 5f–h, yellow curve and Fig. 5i, yellow bars). While this observation was not predicted correctly by the in silico alanine scan, inspection of the structures provides clues. In the PLpro^CoV-2^:hISG15 structure, E70 is

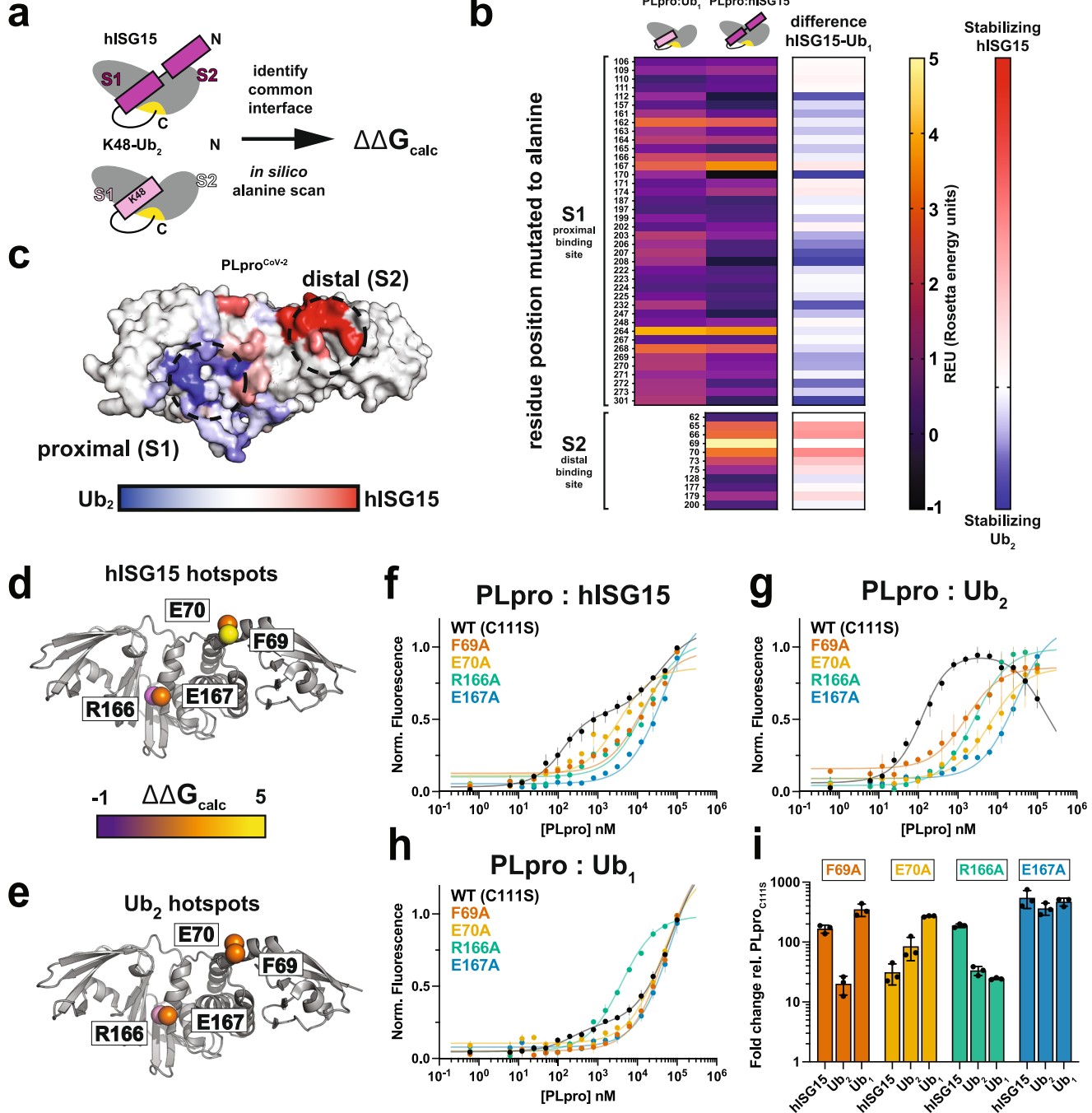

**Fig. 5 | Prediction and validation of specificity-determining surfaces on PLpro^CoV-2.** **a** Schematic illustration of identification of substrate binding surfaces on PLpro^CoV-2 and in silico mutagenesis for heterodimer complexes with hISG15 and K48-Ub₂. **b** Heatmap results of ΔΔG_binding calculations of in silico alanine scan for PLpro^CoV-2 in complex with hISG15 or Ub₁. Interface residue positions in the S1 (proximal) and S2 (distal) binding sites are labeled. The heat map is colored from black to yellow. The last column represents the results of calculations of the difference between REU (Rosetta energy units) for PLpro^CoV-2:hISG15 and PLpro^CoV-2:K48-Ub₂ and is colored from blue to red. **c** Results of in silico mutagenesis for complexes of PLpro^CoV-2 with hISG15 and K48-Ub₂ mapped on the surface representation of PLpro^CoV-2. Hotspot sites identified as those driving stability toward hISG15 are colored red, and those driving stability towards K48-Ub₂ are colored blue. Summary

of PLpro^CoV-2 alanine mutants (F69A, E70A. R166A and E167A) tested for binding to hISG15 (**d**), Ub₂ (**e**), and Ub₁. Mutants are shown as Cα spheres and are colored according to ΔΔG_calc from black to yellow. **f–h** PLpro^CoV-2 WT (C111S) (black), PLpro^CoV-2 F69A (orange), PLpro^CoV-2 E70A (yellow), PLpro^CoV-2 R166A (green), and PLpro^CoV-2 E167A (blue) titrations with hISG15 (**f**), Ub₂ (**g**), and Ub₁ (**h**). Data are shown as triplicates and are plotted as the average with the range of individual replicates. Data were fitted to the preferred 1:2 binding model for WT (C111S) and to the 1:1 binding model for mutants using PALMIST. **i** Summary of fold change in K_ds calculated as a ratio between PLpro^CoV-2 WT (C111S) and PLpro^CoV-2 F69A, PLpro^CoV-2 E70A, PLpro^CoV-2 R166A, PLpro^CoV-2 E167A to hISG15, Ub₂, and Ub₁. Data are shown as triplicates and are plotted as averages with standard deviation. Source data are provided as a Source Data file.

close to S22 but does not form any clear stabilizing interactions (Supplementary Fig. 13c); by contrast, E70 forms a long salt bridge to H68 and K6 in the PLpro$^{CoV-2}$:Ub$_2$ structure that may have been missed by the in silico analysis due to the larger distance (Supplementary Fig. 13c, >3.9 and 5 Å, respectively). Coincidentally, K6 yielded heterogeneous cross-links to acidic residues, including E70, on the thumb domain of PLpro (Fig. 4d), suggesting that charge complementary interactions between the distal Ub and PLpro contribute to the tight binding between PLpro$^{CoV-2}$ and Ub$_2$.

Overall, our combined approach using in silico analysis of the interfaces and binding measurements of mutants uncovered possible means to discriminate binding between PLpro and hISG15 or Ub$_2$, allowing future experiments in more physiological contexts to begin decoupling hISG15 and Ub-dependent effects on virulence and disease. Furthermore, our data suggest that binding of the distal domain of hISG15 and Ub$_2$ to PLpro is determined by different types of interactions (i.e., nonpolar vs charge complementary electrostatics) thus, the association or dissociation rates dictate binding that ultimately manifests as changes in dynamics of the distal Ub (in Ub$_2$) vs distal UBL (in hISG15) (Fig. 6).

## Discussion

The literature highlights that PLpro$^{CoV-1}$ prefers binding to K48-linked polyubiquitin over ISG15[28]. Our work incited by recent studies[26,27] reveals that PLpro$^{CoV-2}$ recognizes human ISG15 and K48-Ub$_2$ with very similar affinities, both in the nanomolar range. Sequence analysis suggests that the PLpro from these two viruses only vary at eight amino acid positions at the substrate binding interface, implicating only minor sequence changes responsible for improving the binding of hISG15. Interestingly, both K48-Ub$_2$ and ISG15 utilize two Ub/UBL domains to recognize and bind PLpro$^{CoV-2}$, but our data show they do it differently. In ISG15, the two UBL domains are connected through a relatively short, likely more rigid peptide linker (DKCDEP in hISG15), and the C-terminal (proximal) UBL binds to S1 site while the N-terminal (distal) UBL binds to S2 site on PLpro$^{CoV-2}$. The amino acid sequences of the proximal and distal UBLs are somewhat different and show distinct contacts that are required for tight binding. In K48-Ub$_2$, two identical Ub units are connected through a flexible linker (RLRGG-K48)[33,34] and contribute differently to the binding. In the high-affinity complex with PLpro, the proximal Ub binds to the S1 site and the distal Ub binds to S2 site. The proximal Ub is very well-ordered and shows multiple interactions with PLpro. The distal Ub interacts with PLpro differently. In the crystal structure, this Ub domain is less ordered, suggesting multiple possible states. NMR data clearly show interactions of the distal Ub with PLpro, but the presence of less occupied states cannot be excluded. K48-Ub$_2$ also binds to PLpro with an altered register where the distal Ub binds to the S1 site while the proximal Ub occupies the S1′ site. This also supports how PLpro disassembles Ub$_3$ by cleaving off unit 3 and the fact that the binding curves of Ub$_2$ and Ub$_1$ to PLpro$^{CoV-2}$ are explained better by the presence of two binding events. Both hISG15 and the K48-Ub$_2$ binding should be sensitive to mutations of S1 and S2 sites in PLpro, but because of different interaction modes,

binding of the hISG15 and the K48-Ub$_2$ substrates is likely to have different sensitivity to such mutations. Furthermore, evolutionary analysis of SARS-CoV-2 variants (Supplementary Fig. 14) highlights sequence variation at key sites, including in the thumb domain, where we have shown that mutations differentially impact ISG15 and Ub$_2$ specificity. Our engineered mutations uncovered nonpolar- vs electrostatics-driven distal UBL (F69) and Ub (E70) contacts in the PLpro$^{CoV-2}$:hISG15 and PLpro$^{CoV-2}$:Ub$_2$ complexes, respectively, that likely underlie the differences in dynamics of the two substrates binding to the protease. These data also suggest that evolutionary variation in the PLpro sequence may alter substrate binding, potentially differentially dysregulating Ub and ISG15 processing, but currently, it is unknown how these alter disease outcomes. Our mutational data suggests that we may be able to engineer mutations that can shift the preference in substrate processing and directly test the contribution of each proteolytic activity in viral pathogenesis.

The structure of PLpro$^{CoV-1}$:K48-Ub$_2$ complex[28] revealed both Ubs bound to the protease S1 and S2 binding sites, with proximal and distal Ubs connected via a non-hydrolyzable triazole linker (in lieu of the native isopeptide linkage) and the C-terminal tail covalently attached to the protease. This may rigidify the K48-Ub$_2$ concealing the true interactions. There is no structure of the ISG15 bound to PLpro$^{CoV-1}$ that could reveal their interactions. Our structure of PLpro$^{CoV-2}$:hISG15 as well as the previous structure of PLpro$^{CoV-2}$:mISG15 reveal a dual UBL domain recognition binding mode despite surprising species sequence variation at the UBL's binding surfaces (Supplementary Fig. 1a–c). Our in silico alanine scan of the interfaces uncovered residues that may play central roles in stabilizing the ISG15 binding mode for PLpro$^{CoV-2}$ and implicated V66 and F69 as being important for stabilizing the distal UBL domain of ISG15.

Much of the focus on understanding how sequence variation impacts pathogenicity, infectivity, and virulence of SARS-CoV-2 has been centered on sequence changes in surface proteins such as the receptor-binding domain (RBD) of the spike protein, which are essential for recognition of ACE2, virus entry into the host cell and thus infectivity. Furthermore, there is concern that mutations in the viral receptors may overcome vaccines which were designed against an engineered prefusion stabilized conformation of RBD of the spike protein, particularly worrisome with emerging variants like the Omicron BA.2 and Ontario WTD clade[41]. Therefore, additional SARS-CoV-2 life cycle steps must be explored, and appropriate key drug targets identified to expand treatment options.

Viral interference with host innate immune response is one of these steps of which ISG15 is integral. Several coronavirus Nsps have been shown to contribute to diminishing this complex response mechanism. Modeling of the protein interfaces suggests that the sequence variation between PLpro from SARS-CoV-1 and SARS-CoV-2 plays a role in the recognition specificity of host factors. Furthermore, we also show that sequence variation within PLpro from 2.3 million SARS-CoV-2 isolates is overall distributed with some hotspots that mimic sequence variation observed between SARS-CoV-1 and SARS-CoV-2 (Supplementary Fig. 14). While we do not

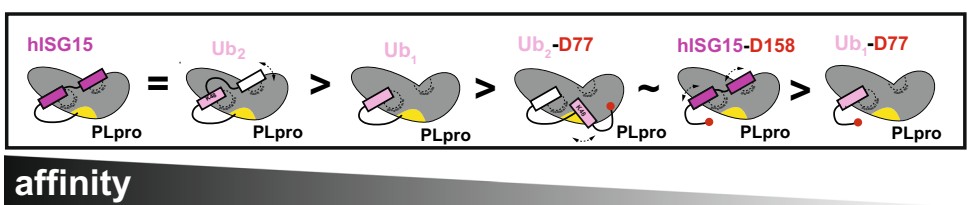

**Fig. 6 | Dual domain-based model for PLpro recognition of K48-linked Ub$_2$ and ISG15.** Schematic representation of differences in binding of PLpro$^{CoV-2}$ with hISG15 and ubiquitin species.

understand how differential recognition of Ub compared to ISG15 impacts pathogenicity and virulence of SARS coronaviruses, a balance between dysregulation of the protective interferon response and ubiquitin-proteasome systems likely influences virus interference with the host defense mechanisms. Future work must be focused on understanding how protease specificity impacts pathogenicity. Furthermore, it remains unknown whether PLpro encodes additional specificity for the substrates that are linked to Ub or ISG15 modifications.

PLpro$^{CoV-2}$ must recognize and process multiple substrates: polyproteins 1a and 1ab, polyUb, and ISG15. It is also known to cleave several other human host proteins. All these substrates have a common sequence recognition motif LXGG; however, they differ in several ways. In coronavirus polyproteins and several host proteins, PLpro cleaves a regular peptide bond. In K48-polyUb and ISG15-modified protein substrates, the cleaved isopeptide bond is between the C-terminal carboxylate of Ub/UBL and a lysine side chain of Ub or other protein, but these latter substrates differ in how the Ubs or UBLs are linked. It is interesting that the conformation of PLpro in complexes with Ub, hISG15, and mISG15 is very similar (0.7 Å rmsd). However, the substrate conformations differ.

What may be the biological implication of single vs dual domain recognition of Ub/UBL for hydrolysis of polyUb/ISG15 modifications? ISG15 is a gene-coded fused dimer; it functions as a di-UBL and is attached covalently to proteins as such. Its specific removal by viral PLpro is also hard-wired to its dimer structure. Ubiquitin is different as it exists as a monomer and is added to a polyUb chain or other proteins in units of monomer. However, PLpro shows the highest affinity for Ub$_2$ (or presumably longer chains) vs Ub$_1$, and it most efficiently removes the proximal Ub (unit 3) from Ub$_3$. Because PLpro binds single Ub less strongly, this suggests that in the cell proteins tagged with polyUb chains containing an odd number of Ubs may accumulate Ub$_1$-substrate adducts, as shown in our cleavage studies.

PLpro$^{CoV-2}$ binds both hISG15 and K48-Ub$_2$ with high and similar affinity but shows weaker interactions with Ub$_1$ or ISG15$_{prox}$/ISG15$_{distal}$. Our data also suggest the presence of lower affinity secondary binding events for hISG15 and K48-Ub$_2$, which can be explained by alternate binding modes. However, the binding mode of these two substrates is quite different. The binding of hISG15 is defined by well-ordered proximal and distal UBLs bound to S1 and to S2 sites, respectively. This may be explained by a combination of a more rigid short (uncleavable) peptide linker between domains and the types of stabilizing interactions between the distal UBL domain and PLpro. In this mode, hISG15 should be cleaved off substrate protein positioned at S1' very efficiently and accumulate free ISG15 at a high viral level. K48-Ub$_2$ binds predominantly using the proximal Ub to the S1 site. The distal Ub binds to the S2 site, but it is less ordered and assumes different states that still contribute to increased affinity. However, in order to be cleaved, the Ub$_2$ substrate must switch to a different, lower-affinity mode with the distal Ub bound to the S1 site and the proximal Ub positioned at the S1' site. Moreover, we noticed that the amino acid sequence of the substrates' tail entering the active site (the LXGG↓X motif) also influenced the rate of cleavage, with acidic residues (E/D) poorly tolerated next to the Gly residue, likely disrupting ionization states of catalytic triad residues (Cys-His-Asp).

In summary, our findings pave the way to understand the interaction of PLpro with hISG15 and (poly)ubiquitin substrates and uncover binding heterogeneity that appears to decouple binding affinity from protease activity. Future experiments will focus on how sequence changes in PLpro can influence the distribution of primary and secondary binding sites of substrates. These experiments will be essential to decouple the different proteolytic activities (Nsps, polyUb, and ISG15) and determine their contribution to viral pathogenesis.

## Methods

### Gene cloning, protein expression, and purification of WT and mutants of PLpro

The gene cloning, protein expression, and purification were performed using protocols published previously in ref. 42. Briefly, the Nsp3 DNA sequence corresponding to PLpro protease of SARS-CoV-2 was optimized for *E. coli* expression using the OptimumGene codon optimization algorithm followed by manual editing and then cloned directly into pMCSG53 vector (Twist Bioscience). PLpro mutants were produced using plasmid amplification with primers encoding the mutation, followed by DpnI cleavage. The plasmids were transformed into the *E. coli* BL21(DE3)-Gold strain (Stratagene). *E. coli* cells harboring plasmids for SARS-CoV-2 PLpro WT and mutants (C111S; C111S,F69A; C111S,E70A; C111S,R166A; C111S,E167A; and C111S,D286N) and ISG15 expression were cultured in LB medium supplemented with ampicillin (150 µg/ml).

For large-scale purification of WT and mutant PLpro$^{CoV-2}$ constructs, 4 L cultures of LB Lennox medium were grown at 37 °C (200 rpm) in the presence of ampicillin 150 µg/ml. Once the cultures reached OD$_{600}$ -1.0, the temperature setting was changed to 4 °C. When the bacterial suspensions cooled down to 18 °C they were supplemented with 0.5 mM IPTG and 40 mM K$_2$HPO$_4$ (final concentration). The temperature was set to 18 °C for 20 h incubation. Bacterial cells were harvested by centrifugation at 7000 × g and cell pellets were resuspended in a 12.5 ml lysis buffer (500 mM NaCl, 5% (v/v) glycerol, 50 mM HEPES pH 8.0, 20 mM imidazole pH 8.0, 10 mM β-mercaptoethanol, 1 µM ZnCl$_2$) per liter culture and sonicated at 120 W for 5 min (4 s ON, 20 s OFF). The cellular debris was removed by centrifugation at 30,000 × g for 90 min at 4 °C. The supernatant was mixed with 3 ml of Ni$^{2+}$ Sepharose (GE Healthcare Life Sciences), which had been equilibrated with lysis buffer supplemented to 50 mM imidazole pH 8.0, and the suspension was applied on Flex-Column (420400-2510) connected to Vac-Man vacuum manifold (Promega). Unbound proteins were washed out via controlled suction with 160 ml of lysis buffer (with 50 mM imidazole pH 8.0). Bound proteins were eluted with 15 ml of lysis buffer supplemented with 500 mM imidazole pH 8.0, followed by Tobacco Etch Virus (TEV) protease treatment at 1:25 protease:protein ratio. The solutions were left at 4 °C overnight. Size exclusion chromatography was performed on a Superdex 75 column equilibrated in lysis buffer. Fractions containing cut protein were collected and applied on a Flex-Column with 3 ml of Ni$^{2+}$ Sepharose which had been equilibrated with lysis buffer. The flow-through and a 7 ml lysis buffer rinse were collected. Lysis buffer was replaced using a 30 kDa MWCO filter (Amicon-Millipore) via 10X concentration/dilution repeated three times to crystallization buffer (20 mM HEPES pH 7.5, 150 mM NaCl, 1 µM ZnCl$_2$, and 10 mM DTT). The final concentration of WT PLpro$^{CoV-2}$ was 25 mg/ml and C111S mutant was 30 mg/ml. For some NMR studies requiring longer measurement times, we also utilized C111S,Y171H PLpro$^{CoV-2}$ variant, which showed similar binding properties to the C111S variant but was more stable in the NMR buffer.

### Expression and purification of unlabeled and isotope-labeled hISG15 and Ub

Human ISG15, as well as genes for the distal (hISG15$_{distal}$) and proximal (hISG15$_{prox}$) UBLs separately, were also synthesized and cloned directly into the pMCSG53 vector. The UBL sequences included amino acids G2–L82 (hISG15$_{distal}$) and L82–G157 (hISG15$_{prox}$) of hISG15. These constructs were purified following the same protocol as for PLpro, except that the buffers did not contain ZnCl$_2$ and a 10 kDa MWCO filter was used for buffer exchange and concentration. The N-terminal polyhistidine tag was removed using TEV protease, which left an additional serine/alanine at the N-terminus, followed by an additional Ni-NTA step. The final concentration of hISG15 was 40 mg/ml. Unlabeled Ub variants were expressed in BL21(DE3) *E. coli* cells containing a helper pJY2 plasmid and purified as described elsewhere[29]. For

expression of uniformly $^{15}$N-labeled Ubs, hISG15, hISG15$_{distal}$, hISG15$_{prox}$, as well as PLpro$^{CoV-2}$ variants, the cells were grown in minimal media containing $^{15}$NH$_4$Cl as the sole source of nitrogen using methods previously described by Varadan et al. [29,31,43].

## Synthesis of K48-polyUb chains

Ub$_2$ and Ub$_3$ chains were assembled from the respective recombinant Ub monomers using controlled chain synthesis catalyzed by Ub-activating E1 enzyme UBA1 and K48-specific Ub-conjugating E2 enzyme UBE2K (aka E2-25K) as detailed elsewhere[29,31,43]. Specifically, Ub variants bearing chain-terminating mutations, Ub-K48R and Ub-D77, were used to ensure that only Ub dimers are produced and to enable incorporation of $^{15}$N-labeled Ub units at the desired distal ($^{15}$N-Ub-K48R) or proximal ($^{15}$N-Ub-D77) position in the resulting chain for NMR and MS studies. D77 was subsequently removed from the proximal Ub by Ub C-terminal hydrolase YUH1. Ub$_3$ chains were made in a stepwise manner, by first removing D77 from the proximal Ub in Ub$_2$ by YUH1 and subsequently conjugating this Ub$_2$ to Ub-D77 using E1 and E2-25K to produce Ub$_3$. The Ub$_3$ chain for MS-based cleavage assays had Ub-K48R at the distal unit, $^{15}$N-labeled Ub at the endo unit, and Ub-D77 at the proximal unit, in order to allow unambiguous identification of the possible cleavage products by mass spectrometry. The correct masses of the synthesized Ub$_2$ and Ub$_3$ chains were confirmed using ESI-MS and SDS-PAGE.

## Microscale thermophoresis binding measurements

MST experiments were performed using NanoTemper Monolith NT.115 available in the Macromolecular Biophysics Resource core at UTSW and the standard protocol was employed during analysis[44]. Ub$_1$, Ub$_1$-D77, K48-Ub$_2$, K48-Ub$_2$-D77, hISG15, hISG15-D158, hISG15$_{distal}$, or hISG15$_{prox}$ were labeled with Cyanine5 NHS ester dye (Cy5) and PLpro$^{CoV-2}$ (C111S; C111S,F69A; C111S,E70A; C111S,R166A, and C111S,E167A mutants) was titrated by a 1:1 serial dilution. Obtained data were fit and analyzed in PALMIST v1.5.8[44,45] using 1:1 and 1:2 binding models and visualized in GUSSI v1.4.2[46] and GraphPad Prism v9. All measurements were done in three replicates. To determine whether the 1:1 or 1:2 binding models were more suitable, we calculated the probability of getting the $\chi^2$ improvement by chance using a t-test. The binding stoichiometry of the complexes was verified using cross-linking of complexes and visualized by SDS-PAGE. PALMIST and GUSSI software is freely available for academic users on the UTSW Macromolecular Biophysics Resource website [https://www.utsouthwestern.edu/research/core-facilities/mbr/software/]. A summary of MST fitting parameters ($K_d$s and errors) for 1:1 and 1:2 binding data are shown in Supplementary Table 1.

## PLpro cleavage assay

K48-Ub$_3$, K48-Ub$_2$, K48-Ub$_2$-D77, and ISG15-Nsp2 (including A158D, A158N, A158E, and A158Q mutants), ISG15-Nsp3 and ISG15-Nsp4 cleavage reactions were performed at 20 °C in 20 mM Tris buffer pH 7.52 containing 100 mM NaCl, 10 mM DTT, and 1 μM ZnCl$_2$. The initial volume was 350 μl and contained 20 μM of Ub$_3$. Upon addition of 0.5 μM PLpro, equal amounts (10 μl) of reaction samples were aliquoted out at given time points, mixed with an equal volume of SDS load buffer, and immediately placed in a water bath at 70 °C for 5 min to stop the reaction. The samples were then loaded onto 15% urea polyacrylamide gel and resolved using SDS-PAGE, and gels were visualized using Gel Doc EZ Imager (Bio-Rad Laboratories). Cleavage of ISG15 fusions was quantified by Image Lab v6.0.1 (Bio-Rad Laboratories) from three independent experiments. For mass spectrometry analyses, cleavage reactions were performed on ice, the samples were buffer exchanged into autoclaved RO water and concentrated to 50 μl volume prior to analysis. The data shown here were obtained on Bruker Maxis-II ultra-high resolution Q-TOF mass spectrometer available at the University of Maryland. Peaks were isolated during separation

and analyzed using MagTran v1.03 b2, samples were analyzed once for every three time points (0, 15, and 25 min).

## Protein crystallization

Crystallizations were performed with the protein-to-matrix ratio of 1:1 using the sitting drop vapor-diffusion method with the help of the Mosquito liquid dispenser (TTP LabTech) in 96-well CrystalQuick plates (Greiner Bio-One). MCSG1, MCSG-2, MCSG3, and MCSG4 (Anatrace), Index (Hampton Research), and Wizard 1&2 (Jena Bioscience) screens were used at 16 °C. The PLpro$^{CoV-2}$-C111S:hISG15 complex (13 mg/ml) crystallized in Index E11 (0.02 M MgCl$_2$, 0.1 M HEPES pH 7.5, 22% (w/v) polyacrylic acid sodium salt 5,100). For the PLpro$^{CoV-2}$-C111S:K48-Ub$_2$-D77 complex (11 mg/ml), crystals appeared in MCSG-2 F11 and were improved in hanging drops with a protein-to-matrix ratio of 2:1 in 0.2 M disodium tartrate, 15% (w/v) PEG3350 after seeding with 1/10 volume of PLpro$^{CoV-2}$-C111S:K48-Ub$_2$-D77 microcrystals. Crystals of the PLpro$^{CoV-2}$-C111S,D286N:K48-Ub$_2$ complex were obtained in 0.2 M disodium tartrate, 15% (w/v) PEG3350 (as above). The Ub$_2$ protein crystallized in Pi-PEG D1 (50 mM acetate buffer pH 4.8, 8.6% PEG 2000 MME, 17.1% PEG 400). The hISG15 protein crystallized in MCSG1 G2 (40 mM potassium phoshate, 16% PEG 8000, and 20% glycerol). Crystals selected for data collection were washed in the crystallization buffer supplemented with 25% glycerol and flash-cooled in liquid nitrogen.

## Data collection, structure determination, and refinement

Single-wavelength X-ray diffraction data were collected at 100 K temperature at the 19-ID beamline of the Structural Biology Center at the Advanced Photon Source at Argonne National Laboratory using the program SBCcollect. The diffraction images were recorded on the PILATUS3 X 6 M detector at 12.662 keV energy (0.9792 Å wavelength) using 0.3° rotation and 0.3 s exposure. The intensities were integrated and scaled with the HKL3000 suite v720[47]. Intensities were converted to structure factor amplitudes in the truncate v1.17.29 program[48] from the CCP4 v7.1.013 package[49]. The structures were determined by molecular replacement using HKL3000 suite v720 incorporating the program MOLREP v11.7.02[50–52]. The coordinates of PLpro$^{CoV-2}$ in complex with ubiquitin propargylamide (PDB ID: 6XAA) and PLpro$^{CoV-2}$-C111S with mISG15 (PDB ID: 6YVA) were used as the starting models for PLpro$^{CoV-2}$-C111S:K48-Ub$_2$ and PLpro$^{CoV-2}$-C111S:hISG15 structure solutions, respectively. For Ub$_2$ and hISG15 proteins, the structures of PLpro$^{CoV-1}$ bound to a K48-Ub$_2$ activity-based probe (PDB ID: 5E6J) and ISG15 (PDB ID: 1Z2M) were used as the starting models. The initial solutions were refined, both rigid-body refinement and regular restrained refinement by REFMAC v5.8.0267 program[53] as a part of HKL3000 v720[47]. Several rounds of manual adjustments of structure models using COOT v0.7.2[54] and refinements with REFMAC program[53] from CCP4 suite[49] were done. The stereochemistry of the structure was validated with PHENIX v1.19-4092 suite[55] incorporating MOLPROBITY v4.02b-467[56] tools. Images were created using PyMOL v1.8.4.2 & v2.5.2. A summary of data collection and refinement statistics is given in Supplementary Table 2.

## NMR data collection and analysis

NMR measurements were performed at 25 °C on Avance III 600 and 800 MHz Bruker NMR spectrometers equipped with cryoprobes. The data were processed using Topspin (Bruker) and analyzed using Sparky v3.190[57]. NMR signal assignments for hISG15 were obtained from Biological Magnetic Resonance Data Bank (BMRB Entry ID 5658), assignments for Ub$_1$ were based on BMRB Entry ID 17769, and for Ub$_2$ on BMRB Entry IDs 30602 and 19406, and adjusted to the temperature and buffer conditions used in our studies. NMR signal assignments for PLpro$^{CoV-2}$ are currently unavailable. Assignment of NMR signals of this 36 kDa protein is a challenging task that is further severely complicated by the poor stability (hours) of the isolated PLpro in the solution.

However, we were able to identify and confirm by mutagenesis the unique peaks observed in the spectral regions characteristic for indole NH tryptophan and for imidazole NH histidine signals, as belonging to the only two tryptophans (W93 and W106) in PLpro[CoV-2] and the active site histidine H272 (Fig. 3 and Supplementary Figs. 8, 10).

The protein samples for NMR measurements were prepared in 50 mM HEPES buffer or in 20 mM Tris buffer, both at pH 7.42 and containing 100 mM NaCl, 1 mM TCEP, 1 μM $ZnCl_2$, 0.2% (w/v) $NaN_3$, and 10% (v/v) $D_2O$. Binding studies by NMR were carried out by adding pre-calculated amounts of unlabeled PLpro[CoV-2]-C111S to [15]N-labeled hISG15 or K48-Ub$_2$ (with either distal or proximal Ub [15]N-labeled) up to ~1:1 molar ratio, or 2:1 molar ratio to Ub$_1$ and monitoring changes in 2D [1]H-[15]N SOFAST-HMQC and/or [1]H-[15]N TROSY as well as 1D [1]H spectra. The initial binding studies were performed in HEPES buffer and subsequently repeated in Tris buffer, both produced similar results. Reciprocal titrations were performed by adding unlabeled Ub$_2$, Ub$_1$, or hISG15 (or D158-extended) to [15]N-labeled PLpro[CoV-2]-C111S or PLpro[CoV-2]-C111S,Y171H in a 1.5:1 (Ub$_2$, hISG15:PLpro) or up to 8:1 (Ub$_1$:PLpro) molar ratio. [15]N-labeled PLpro[CoV-2]-C111S,Y171H was primarily used for lengthy TROSY experiments as this variant proved to be more stable than PLpro[CoV-2]-C111S in the NMR buffer. Both PLpro[CoV-2] variants had very similar NMR spectra and essentially identical signal perturbations upon binding of all the substrates tested (see Supplementary Fig. 8).

The protein concentrations in NMR studies in Tris buffer were as follows: 150:150 μM for hISG15:PLpro[CoV-2], 180:270 μM for hISG15-D158:PLpro[CoV-2], 83:104 μM for [15]N-distal Ub$_2$:PLpro[CoV-2], 191:240 μM for [15]N-proximal Ub$_2$:PLpro[CoV-2], 153:306 μM for Ub$_1$:PLpro[CoV-2], 125:250 μM for Ub$_1$-D77:PLpro[CoV-2]. Experiments with [15]N-PLpro[CoV-2] used 301.5:201.5 μM hISG15:PLpro[CoV-2], 225:150 μM hISG15-D158:PLpro[CoV-2], 137.5:110 μM Ub$_2$:PLpro[CoV-2], 404:202 μM Ub$_2$-D77:PLpro[CoV-2], up to 1140:141 μM Ub$_1$:PLpro[CoV-2], 1000:125 μM Ub$_1$-D77:PLpro[CoV-2]. For NMR measurements in HEPES buffer, the concentrations were 115:115 μM for hISG15:PLpro[CoV-2]; 71:142 μM for [15]N-distal Ub$_2$:PLpro[CoV-2]; 75:150 μM for [15]N-proximal Ub$_2$: PLpro[CoV-2]; and 81:151 μM for Ub$_1$:PLpro[CoV-2]. Results of NMR experiments of the competition assay and PLpro[CoV-2] cleavage of Ub$_3$ were plotted using OriginPro v8.

**Cross-linking mass spectrometry analysis**

Our group has developed standardized protocols for cross-linking and data analysis of samples. For complexes between PLpro[CoV-2] and Ub$_1$, Ub$_1$-D77, K48-Ub$_2$, K48-Ub$_2$-D77, or hISG15, we incubated the protease with the substrate at a 1:4 molar ratio for 1 h at 25 °C. For disuccinimidyl suberate (DSS)[58] reactions, samples were cross-linked with a final 1 mM DSS (DSS-d$_0$ and -d$_{12}$, Creative Molecules) for three minutes at 37 °C while shaking at 350 rpm. For sulfonyl fluoride (SuFEx)[59] reactions, samples were cross-linked with a final 0.658 mM SuFEx (a kind gift from William DeGrado, UCSF) for 1 hour at 37 °C while shaking at 350 rpm. For DMTMM[38] reactions, samples were cross-linked with a final 43 mM DMTMM (Sigma-Aldrich) for 15 min at 37 °C while shaking at 600 rpm. All cross-linking reactions were quenched with 172 mM (four times excess) ammonium bicarbonate for 30 min at 37 °C while shaking at 350 rpm. Samples were resolved on SDS-PAGE gels (NuPAGE™, 4 to 12%, Bis-Tris, 1.5 mm). For glutaraldehyde cross-linking, we used 1:4, 1:1 and 1:0.25 ratios of PLpro:substrate using 6 μM protease in all reactions. Samples were preincubated at 25 °C for 15 min followed by the addition of 0.05% glutaraldehyde (Sigma) for 1 min and quenched with Tris pH 8.0 to a final concentration of 0.2 M. Cross-linked species and control reactions were resolved by SDS-PAGE.

For a set of samples cross-linked with DMTMM for XL-MS analysis, bands corresponding to PLpro[CoV-2]:substrate heterodimers were extracted from the gel using the following protocol. Bands were cut into 3–4 mm pieces per individual tube and washed twice with water, and then covered in acetonitrile/50 mM $NH_4HCO_3$ (mixed in ratio 2:3, v/v) and sonicated for 5 min three times, with removing the supernatant after each time. Gel pieces were incubated for 5 min at RT with

100% acetonitrile until they became white, and then were lyophilized. Pieces were then incubated in 25 mM $NH_4HCO_3$ with 10 mM DTT for 1 h at 56 °C while shaking at 350 rpm. Samples were allowed to cool down, the supernatant was removed, and gel pieces were incubated in 25 mM $NH_4HCO_3$ with 55 mM iodoacetamide for 40 min at RT in the dark. Then, gel pieces were washed sequentially with 25 mM $NH_4HCO_3$, 50% acetonitrile, and lastly, 100% acetonitrile with quick vortexing after each wash. The supernatant was then removed and gel pieces were lyophilized for 5 min followed by trypsin digestion where 1:50 (m/m) trypsin (Promega) in 50 mM ammonium bicarbonate was added and samples were incubated overnight at 37 °C while shaking at 600 rpm. 2% (v/v) formic acid was added to acidify the reaction and the supernatants were further purified by reversed-phase Sep-Pak tC18 cartridges (Waters), next flash frozen in liquid nitrogen and lyophilized. The dried samples were resuspended in water/acetonitrile/formic acid (95:5:0.1, v/v/v) to a final concentration of ~0.5 μg/μl. About 2 μl of each was injected into Eksigent 1D-NanoLC-Ultra HPLC system coupled to a Thermo Orbitrap Fusion Tribrid system at the UTSW Proteomics core.

The mass spectrometry data were analyzed by in-house version of xQuest 2.1.5 pipeline[60]. Thermo RAW data files were first converted to open.mzXML format using msconvert (proteowizard.sourceforge.net). The mass spectra across replicates yielded similar intensities. Search parameters were set based on DMTMM as the cross-linking reagent as follows: maximum number of missed cleavages = 2, peptide length = 5–50 residues, fixed modifications = carbamidomethyl-Cys (mass shift = 57.02146 Da), variable modification = oxidation of methionine (mass shift = 15.99491 Da), mass shift of cross-linker = −18.010595 Da, no monolink mass specified, MS1 tolerance = 15 ppm, and MS2 tolerance = 0.2 Da for common ions and 0.3 Da for cross-link ions; search in enumeration mode. Next, in-house shell script was employed to identify cross-links between lysines and acidic residues. FDRs were estimated by xProphet as a part of the same version of xQuest[61] to be 9.8–77.8%. For each experiment, five replicate datasets were compared and only cross-link pairs that appeared in all datasets (PLpro[CoV-2]:K48-Ub$_2$ and PLpro[CoV-2]:Ub$_1$) or at least in four datasets (PLpro[CoV-2]:hISG15, PLpro[CoV-2]:K48-Ub$_2$-D77) were used to generate a consensus dataset. Cross-linking mass spectrometry data are available in Source Data file provided with this study.

**Modeling of alternate Ub binding sites on PLpro[CoV-2]**

To build an ensemble of alternate Ub binding sites on PLpro[CoV-2] outside of the canonical proximal domain on the S1 site, we employed docking procedure that combined a geometric restraint between K48 of the immobile proximal Ub to the C-terminus of the mobile Ub to sample alternate S2' binding sites. In the alternative scenario, we employed a geometric restraint between C-terminus of the immobile proximal Ub to K48 of the mobile Ub to sample alternate S1' binding sites below the active site. An initial conformation of the PLpro[CoV-2] bound to Ub in the proximal site was built from our structure (PDB ID: 7RBR) and converted into a single chain. We next added a mobile Ub as a second chain and produced over 5000 low-resolution centroid mode models employing two different geometric restraints that sampled alternate S2' and S1' binding sites. Each structure was minimized and the total energy of the PLpro[CoV-2]:Ub$_2$ complexes was plotted as a function of a sum distances of Cβ-Cβ from the experimental cross-links using an in-house script. All simulations were performed with RosettaDock protocol[62] as a part of Rosetta v3.13 suite and ran on UTSW's BioHPC computing cluster. All plots were generated with GraphPad Prism v9. Images were created using PyMOL v1.8.4.2 & v2.5.2.

**Energetic analysis of PLpro[CoV-2] in complex with hISG15 and K48-Ub$_2$**

Models of PLpro[CoV-2] bound to two different substrates, hISG15 and K48-Ub$_2$, that were used in the subsequent in silico alanine scan were prepared as follows. For the PLpro[CoV-2]:hISG15 complex, we used a

heterodimer conformation derived from our crystal structure (PDB ID: 7RBS). To create the complex between PLpro[CoV-2] and K48-Ub$_2$, we used the conformation and binding mode of K48-Ub$_2$ bound to PLpro[CoV-1] (PDB ID: 5E6J as a template[28]. Briefly, our PLpro[CoV-2] (PDB ID: 7UV5) was aligned to PLpro[CoV-1] bound to K48-Ub$_2$ to produce a tentative model of PLpro[CoV-2]:K48-Ub$_2$. As a control, a model of PLpro[CoV-2] with single proximal Ub visible in our density was also analyzed. Next, we applied a relax protocol in Rosetta for both complexes: PLpro[CoV-2]:K48-Ub$_2$ and PLpro[CoV-2]:hISG15. To guarantee that each instance of relax is being run with different randomizations, groups of nstruct were run with different, randomly generated seeds using random.org. From 100 total structures (4 × 25 nstruct for computational efficiency) for each heterocomplex, the lowest energy structure was identified and used in further steps. The list of PLpro residues that may be engaged in interacting with its substrate was created by identifying PLpro residues within 4.0 Å of either hISG15 or K48-Ub$_2$. The union list of interacting residues identified with heterocomplexes PLpro[CoV-2]:hISG15 and PLpro[CoV-2]: K48-Ub$_2$ was used in the next step. For PLpro[CoV-2] in complex with hISG15 or K48-Ub$_2$ 51 positions were used to describe the combined interface. Selection of common interface residues was carried out in PyMOL v2.5.2. Flex ΔΔG protocol was used as described previously[39]. The code is available on Kortemme Lab GitHub [https://github.com/Kortemme-Lab]. Briefly, selected interacting residues were mutated to alanines. Parameters were used (all default settings): nstruct = 35, max_minimization_iter = 5000, abs_score_convergence thresh = 1.0, number_backrub_trials = 35000, and to enable earlier time points backrub_trajectory_stride was set to 7000. The ΔΔG$_{binding}$ score for the last iteration is shown in the Results. These simulations were performed using BioHPC computing cluster at UT Southwestern Medical Center. The results, in raw REU (Rosetta energy units), are shown as a heatmap with ΔΔG$_{binding}$ values but also as the difference between the ΔΔG$_{binding}$ from hISG15 compared to K48-Ub$_2$. The plots were made using GraphPad Prism v9 and mapped onto the protease structure using PyMOL v1.8.4.2 and v2.5.2. The relax protocol and Flex ΔΔG used Rosetta v3.13 and v3.12, respectively.

### Sequence comparison of Ub, ISG15, and sequence variation across PLpro[CoV-2] in SARS-CoV-2

Alignments were produced in Clustalo[63] and visualized in Seaview[64]. Sequence identity between Ub$_2$, hISG15, and mISG15 was calculated using Blast[65]. PLpro[CoV-2] sequence variation from 2.3 million sequences (as of October 18, 2021) was derived from the coronavirus3D database[66]. The per residue mutational frequencies were mapped onto a PLpro[CoV-2] structure in the context of a bound K48-Ub$_2$ or hISG15.

### Data availability

The structural datasets generated during the current study are available in the Protein Data Bank repository [https://www.rcsb.org/] under accession codes PDB ID: 7RBR for PLpro[CoV-2], C111S mutant, in complex with K48-Ub$_2$; PDB ID: 7RBS for PLpro[CoV-2], C111S mutant, in complex with human ISG15; PDB ID: 7UV5 for PLpro[CoV-2], C111S,D286N mutant, in complex with K48-Ub$_2$; PDB ID: 7S6P for human ISG15 alone, and PDB ID: 7S6O for K48-Ub$_2$ alone. Diffraction images are available on Integrated Resource for Reproducibility in Macromolecular Crystallography repository set up by W. Minor laboratory [https://proteindiffraction.org] for PDB IDs 7RBR; 7RBS; 7UV5; 7S6P; 7S6O. NMR signal assignments for hISG15 used in this study are available under BMRB Entry ID 5658, assignments for Ub$_1$ were based on BMRB Entry ID 17769, and for Ub$_2$ on BMRB Entry IDs 30602 and 19406. All MST, processed cross-linking mass spectrometry, and ΔΔG$_{calc}$ data are available in Source Data provided with this paper. Docking models used in this study are available on Zenodo under accession number 7768418. Raw MS data used for the XL-MS analysis are available in the MassIVE database under the accession number MSV000091075 and in the ProteomeXchange under the identifier PXD040822. Source data are provided with this paper.

### Code availability

All ΔΔG$_{calc}$ calculations were performed using the flex_ddg protocol that is available on Kortemme Lab GitHub [https://github.com/Kortemme-Lab/flex_ddG_tutorial] using Rosetta v3.12 [https://www.rosettacommons.org/].

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

## Acknowledgements

We thank the members of the SBC at Argonne National Laboratory, especially Darren Sherrell, Krzysztof Lazarski, and Alex Lavens, for their help with setting beamline and data collection at beamline 19-ID. We thank Chad Brautigam in the UTSW Macromolecular Biophysics Resource for help with MST data analysis. We also appreciate the help of the Proteomics Core Facility at UTSW. Funding for this project was provided in part by federal funds from the National Institute of Allergy and Infectious Diseases, National Institutes of Health, Department of Health and Human Services, under Contracts HHSN272201700060C and 75N93022C00035 and by the DOE Office of Science through the National Virtual Biotechnology Laboratory, a consortium of DOE national laboratories focused on response to COVID-19, with funding provided by the Coronavirus CARES Act. Results shown in this report are derived from work performed at SBC funded by the U.S. Department of Energy, Office of Biological and Environmental Research and operated for the DOE Office of Science at the Advanced Photon Source by Argonne National Laboratory under Contract No. DE-AC02-06CH11357. This research was supported by NIH grant GM065334 to D.F. NMR experiments were performed at UMD on instruments supported in part by NSF grant DBI1040158, and the instrument for mass spectrometry measurements was supported by NSF grant CHE-2018860. We acknowledge Yue Li at UMD for help with Maxis-II Q-TOF measurements. E.E. was supported by Undergraduate Maryland Summer Scholarship. L.A.J. is supported by an Effie Marie Cain Scholarship in Medical Research. P.M.W. is supported by an O'Donnell Brain Institute Pilot grant.

## Author contributions

A.J., L.A.J., K.F., and D.F. initiated the project. R.J. cloned, expressed, and purified the first batch of protein. M.E. continued with different constructs, mutations, and complexes. C.T. purified and crystallized protein and complexes with ligands, and with J.O. collected diffraction data. J.O. determined, refined, and, together with K.M., analyzed structures. B.T.L. and E.E. synthesized all Ub chains, expressed and purified all proteins for NMR studies, and carried out NMR studies. B.T.L. and C.T. performed $Ub_3$ cleavage reactions and their MS analysis. B.T.L. performed $Ub_2$ cleavage reactions and inhibition assays as well as NMR studies of $Ub_3$ cleavage reactions and $Ub_2$/ISG15 competition for PLpro binding. B.T.L. and D.F. analyzed NMR data. C.T. performed ISG15-Nsp cleavage reactions and analysis. P.M.W. carried out all the MST binding, XL-MS and docking experiments, and analysis. ΔΔG calculations and analysis were carried out by P.M.W. and V.M. Finally, D.F., A.J., and L.A.J. conceived and directed the research as well as wrote the manuscript.

## Competing interests

The authors declare no competing interests.
