## [Peer Review File · Nature Communications]

REVIEWERS' COMMENTS

Reviewer #1 (Remarks to the Author):

I am satisfied with the authors' response to my previous review. The experimental testing of the in silico alanine-scanning has yielded further insights into the differences in binding to each of the substrates, and the extra crystal structure goes well with the dynamic analyses.

Reviewer #2 (Remarks to the Author):

My comments have been addressed appropriately.

Reviewer #3 (Remarks to the Author):

As outlined in my initial review, this is clearly an important paper that should be published in a high quality journal. The authors demonstrate that PLpro of CoV-2 binds both hISG15 and K48-Ub2 with high and similar affinities, but shows weaker interactions with Ub1 suggesting that additional interactions outside of Ub1 stabilize the complex. Their data also suggest the presence of lower affinity secondary binding sites for hISG15 and K48-Ub2, which appear to be due to not-yet-characterized alternate binding modes. In particular, binding thermodynamics studies with various Ub constructs and hydrolysis kinetics with triUb substrates reveal complex binding modes that have not been previously recognized. Using multiple experimental and in silico data, the authors confirm that the high affinity binding to ISG15 is determined by dual domain recognition, while K48-Ub2 is recognized mainly through proximal Ub domain. I strongly support publishing this work as soon as possible

My principal concern is the rigor of the NMR studies. Despite the author's rebuttals, the interpretation of these data would be much stronger if backbone resonance assignments were available for PLpro. I fully expected the authors to come back with at least a partial set of backbone resonance assignments for unbound PLpro to guide the interpretation of the NMR data. However, I appreciate that PLpro stability issues may preclude determination of backbone resonance assignments in a time frame short enough to complete the publication of this otherwise very strong paper.

Perhaps the fact that significant challenges were encountered in obtaining backbone resonance assignments should be mentioned in the discussion.

This is not to say that the NMR studies are problematic or misinterpreted. The NMR studies using isotope-labeled ISG15 and Ub constructs provide valuable data that complement the other studies outlined in the paper. The limited data on Trp106 and His272 of PLpro allow important interpretations of data in terms of binding mechanisms, and the NMR-based UB3 cleavage data (Suppl Fig S3) provide important data about the cleavage process.

I also concur with the authors that while the work does not address the biological significance of the differences in the structures of these complexes, the study does reveal important differences in Ub2 and ISG15 recognition which should be published with high urgency.

Minor point - the roles of every author should be stated in the Author Contributions

Reviewer #1 (Remarks to the Author):

I am satisfied with the authors' response to my previous review. The experimental testing of the in silico alanine-scanning has yielded further insights into the differences in binding to each of the substrates, and the extra crystal structure goes well with the dynamic analyses.

We want to thank the Reviewer for their positive assessment of our study.

Reviewer #2 (Remarks to the Author):

My comments have been addressed appropriately.

We want to thank the Reviewer for their positive assessment of our study.

Reviewer #3 (Remarks to the Author):

As outlined in my initial review, this is clearly an important paper that should be published in a high quality journal. The authors demonstrate that PLpro of CoV-2 binds both hISG15 and K48-Ub2 with high and similar affinities, but shows weaker interactions with Ub1 suggesting that additional interactions outside of Ub1 stabilize the complex. Their data also suggest the presence of lower affinity secondary binding sites for hISG15 and K48-Ub2, which appear to be due to not-yet-characterized alternate binding modes. In particular, binding thermodynamics studies with various Ub constructs and hydrolysis kinetics with triUb substrates reveal complex binding modes that have not been previously recognized. Using multiple experimental and in silico data, the authors confirm that the high affinity binding to ISG15 is determined by dual domain recognition, while K48-Ub2 is recognized mainly through proximal Ub domain. I strongly support publishing this work as soon as possible

My principal concern is the rigor of the NMR studies. Despite the author's rebuttals, the interpretation of these data would be much stronger if backbone resonance assignments were available for PLpro. I fully expected the authors to come back with at least a partial set of backbone resonance assignments for unbound PLpro to guide the interpretation of the NMR data. However, I appreciate that PLpro stability issues may preclude determination of backbone resonance assignments in a time frame short enough to complete the publication of this otherwise very strong paper.

Perhaps the fact that significant challenges were encountered in obtaining backbone resonance assignments should be mentioned in the discussion.

This is not to say that the NMR studies are problematic or misinterpreted. The NMR studies using isotope-labeled ISG15 and Ub constructs provide valuable data that complement the other studies outlined in the paper. The limited data on

Trp106 and His272 of PLpro allow important interpretations of data in terms of binding mechanisms, and the NMR-based UB3 cleavage data (Suppl Fig S3) provide important data about the cleavage process.

I also concur with the authors that while the work does not address the biological significance of the differences in the structures of these complexes, the study does reveal important differences in Ub2 and ISG15 recognition which should be published with high urgency.

We want to thank the Reviewer for their comment on NMR. As suggested, we have added an explanation of challenges in obtaining backbone resonance assignments for PLpro in the Methods.

Minor point - the roles of every author should be stated in the Author Contributions

We want to thank the Reviewer for pointing this out. We have corrected the Author Contributions section accordingly.